# Place fields of single spikes in hippocampus involve Kcnq3 channel-dependent entrainment of complex spike bursts

Xiaojie Gao[1,2,8], Franziska Bender[1,2,8], Heun Soh[3], Changwan Chen[4,5], Mahsa Altafi[6], Sebastian Schütze[1,7], Matthias Heidenreich[1,7], Maria Gorbati[1,2], Mihaela-Anca Corbu [4], Marta Carus-Cadavieco[1,2], Tatiana Korotkova[1,2,4,5], Anastasios V. Tzingounis[3], Thomas J. Jentsch [1,2,7✉] & Alexey Ponomarenko [1,2,6✉]

Hippocampal pyramidal cells encode an animal's location by single action potentials and complex spike bursts. These elementary signals are believed to play distinct roles in memory consolidation. The timing of single spikes and bursts is determined by intrinsic excitability and theta oscillations (5–10 Hz). Yet contributions of these dynamics to place fields remain elusive due to the lack of methods for specific modification of burst discharge. In mice lacking Kcnq3-containing M-type K$^+$ channels, we find that pyramidal cell bursts are less coordinated by the theta rhythm than in controls during spatial navigation, but not alert immobility. Less modulated bursts are followed by an intact post-burst pause of single spike firing, resulting in a temporal discoordination of network oscillatory and intrinsic excitability. Place fields of single spikes in one- and two-dimensional environments are smaller in the mutant. Optogenetic manipulations of upstream signals reveal that neither medial septal GABA-ergic nor cholinergic inputs alone, but rather their joint activity, is required for entrainment of bursts. Our results suggest that altered representations by bursts and single spikes may contribute to deficits underlying cognitive disabilities associated with *KCNQ3*-mutations in humans.

[1] Leibniz-Forschungsinstitut für Molekulare Pharmakologie (FMP), Berlin, Germany. [2] NeuroCure Cluster of Excellence, Charité Universitätsmedizin, Berlin, Germany. [3] University of Connecticut, Storrs, CT, USA. [4] Max Planck Institute for Metabolism Research, Cologne, Germany. [5] Institute for Vegetative Physiology, Faculty of Medicine, University of Cologne, Cologne, Germany. [6] Institute of Physiology and Pathophysiology, Friedrich-Alexander-Universität Erlangen-Nürnberg, Erlangen, Germany. [7] Max-Delbrück-Centrum für Molekulare Medizin (MDC), Berlin, Germany. [8] These authors contributed equally: Xiaojie Gao, Franziska Bender. ✉email: jentsch@fmp-berlin.de; alexey.ponomarenko@fau.de

How is the balance between reliability and precision of neuronal communication regulated in the brain? Hippocampal pyramidal cells fire about half of their action potentials in complex spike bursts[1]—several spikes with accommodating amplitudes and interspike intervals (ISI) of 3–15 ms. Bursts increase the probability of transmitter release, leading to more reliable postsynaptic responses than single spikes[2], yet are less informative about the timing of presynaptic activity. Single spikes and bursts provide therefore complementary trade-offs between precision and reliability of information transfer[3]. In the CA1 area, cells participating in engrams of new experiences fire more bursts and encode less precise spatial representations, in contrast to robust spatial coding by neurons with a higher firing probability of single spikes[4], genetic suppression of which does not interfere with hippocampus-dependent learning[3].

Firing of place cells in the dorsal CA1 area[5] appears to be driven by signals of self-motion and spatial cues from entorhinal cortex grid and border cells, respectively[6–8]. The firing of upstream CA3 place cells is more experience-dependent and has been shown to encode substantially different environments[9]. Subcortical afferents, including those from the midline thalamus, the medial septum (MS), and aminergic neurons signal to hippocampus information about sensory stimuli relevant for learning and goal-directed behaviors[10–12]. The translation of input signals from cortical regions into location-specific hippocampal output involves subcortical regulation of excitability in the CA1 area by theta rhythm, activity-dependent plasticity, local inhibition, and intrinsic excitability[13–16].

A slowly activating and non-inactivating potassium conductance, mediated by heteromeric Kcnq2/Kcnq 3 (Kv7.2/Kv7.3) and Kcnq5/Kcnq3 (Kv7.5/Kv7.3) voltage-gated $K^+$ channels, contributes to resting membrane potential in hippocampal pyramidal cells[17]. These ‚M-type' channels can be inhibited by acetylcholine via M1 receptors[18,19], which promote repetitive firing of pyramidal cells[18,20]. Together with Kcnq2, Kcnq3 subunits are targeted to axon initial segments. Here Kcnq3-containing channels regulate the functional availability of $Na^+$ channels and affect spontaneous spiking, action potential amplitude, and propagation[17,21,22]. Kcnq/M-currents in somata and dendrites contribute to medium after-hyperpolarization, reduce excitability, prolong interspike intervals (ISIs) and regulate synaptic integration, and subthreshold resonance[22–26]. In contrast to Kcnq5, which modulates hippocampal information processing by shunting inhibitory postsynaptic currents in the CA3 area[27], Kcnq3 is strongly expressed also in the CA1 area[28]. Mutations in the genes encoding either subunit of heteromeric KCNQ2/3 voltage-gated potassium channels have been linked to childhood epilepsy[29–32]. Recently, cognitive impairments including autism spectrum disorders and intellectual disability associated with recurrent seizures have been associated with mutations and polymorphisms of the human KCNQ3 gene[33–37]. Yet the significance of the Kcnq3 subunit for cognitive information processing in vivo remains unknown.

Here, capitalizing on a crucial position of Kcnq3 in the control of high-frequency discharge of pyramidal cells, we used constitutive as well as pyramidal cell-specific Kcnq3 gene ablation along with electrophysiological and optogenetic approaches to study cellular, network and circuit mechanisms of spatial representations by bursts and single action potentials. We found specific effects of Kcnq3 on burst discharge of pyramidal cells ex vivo and in vivo, and studied the timing of bursts in cells lacking Kcnq3 during spatial navigation and alert immobility. These behavioral states differ in levels of cholinergic stimulation provided to hippocampus by MS afferents. Changes of bursts timing during theta oscillations were accompanied by altered representations of animal's position by single spikes and could be reproduced by optogenetic manipulations of MS afferents. These findings highlight the role of theta oscillations in the cholinergic modulation of hippocampal representations[38] and demonstrate a mechanism for the interaction of coding by bursts and single spikes.

## Results

**Kcnq3-containing M-channels attenuate burst discharge of pyramidal cells.** To investigate the impact of burst discharge on the information processing in the CA1 area we generated $Kcnq3^{-/-}$ mice[39] (Supplementary Fig. 1a, b). Disruption of Kcnq3 abolished the expression of both the Kcnq3 mRNA (Supplementary Fig. 1c) and the Kcnq3 protein (Supplementary Fig. 1d), but did not lead to a compensatory upregulation of Kcnq2 expression (Supplementary Fig. 1c). Hence, $Kcnq3^{-/-}$ mice likely form increased levels of Kcnq2 homomeric at the expense of Kcnq2/3 heteromeric channels. This is expected to result in a large decrease of M-current magnitude since currents through heteromeric Kcnq2/3 channels are much larger than those mediated by homomeric Kcnq2 channels[40,41] (Fig. 1a,b). Furthermore, homomeric Kcnq2 channels are less modulated by M1-receptors, which act by decreasing levels of phosphatidylinositol 4,5-bisphosphate that binds preferentially to Kcnq3[42,43]. Ex vivo, $Kcnq3^{-/-}$ and $Kcnq3^{+/+}$ pyramidal cells did not differ in their resting membrane potential and input resistance (Table 1, Supplementary Fig. 2), possibly because homomeric Kcnq2 channels that remain in $Kcnq3^{-/-}$ neurons prevent a significant change of these parameters. They also displayed similar amplitudes of small conductance calcium-activated potassium (SK-) channel independent (apamin-insensitive) medium after-hyperpolarization (mAHP, Supplementary Fig. 2), in agreement with previous reports[24,26]. Mutant and control pyramidal cells fired action potentials of a similar amplitude and with a similar rate of depolarization (Table 1). However, the rate of repolarization was higher and action potentials were shorter in the mutant (Table 1). In response to the same amount of injected current, $Kcnq3^{-/-}$ pyramidal cells fired more repetitive action potentials than controls (Fig. 1c, d). Furthermore, trains of action potentials evoked by current injections displayed overall higher frequencies in mutant cells (Fig. 1e). Accordingly, pyramidal cells in vivo spontaneously fired bursts which mostly consisted of a larger number of spikes in $Kcnq3^{-/-}$ than in control mice (Fig. 1f, g). Surprisingly, the rate of burst occurrence did not differ between genotypes ($Kcnq3^{+/+}$, 0.23 ± 0.01 Hz, n = 306 cells from 6 mice; $Kcnq3^{-/-}$, 0.24 ± 0.01 Hz, n = 413 cells from 4 mice; p = 0.08, Mann–Whitney-U-Test), but the rate of single spikes and, consequently, the average firing rate of pyramidal cells were reduced in the mutant (single spikes: $Kcnq3^{+/+}$, 0.80 ± 0.03 Hz, n = 309 cells; $Kcnq3^{-/-}$, 0.49 ± 0.02 Hz, n = 413 cells; p < 0.0001; all spikes: $Kcnq3^{+/+}$, 1.31 ± 0.04 Hz; $Kcnq3^{-/-}$, 1.12 ± 0.03 Hz; p < 0.0001, Mann–Whitney-U-Test).

In control mice the initial three spikes of either short or long bursts were emitted in vivo with progressively increasing ISIs, resulting in frequency accommodation by ~20 Hz (Fig. 1f,h). In $Kcnq3^{-/-}$ mice, frequency accommodation was dramatically diminished (Fig. 1f,h,i). Furthermore, bursts in the mutant had overall higher frequency than in the control (interspike intervals: $Kcnq3^{+/+}$, 7.42 ± 0.04 ms, n = 308 cells; $Kcnq3^{-/-}$, 6.67 ± 0.02 ms, n = 413 cells; $F_{1,7604}$ = 195.2, p < 0.0001, ANOVA).

The more prominent burst firing in $Kcnq3^{-/-}$ pyramidal cells was accompanied by a reversed relationship between burst length and intraburst frequency. In contrast to controls, in the mutant, longer bursts had higher frequency of spikes (i.e., shorter ISIs) than shorter bursts (Fig. 1f, h, i). Hence Kcnq3-containing channels attenuate high-frequency firing of pyramidal cells and influence spike timing during complex spike bursts.

**Kcnq3 loss alters timing of complex spike bursts during spatial navigation.** Kcnq3-containing channels may influence the

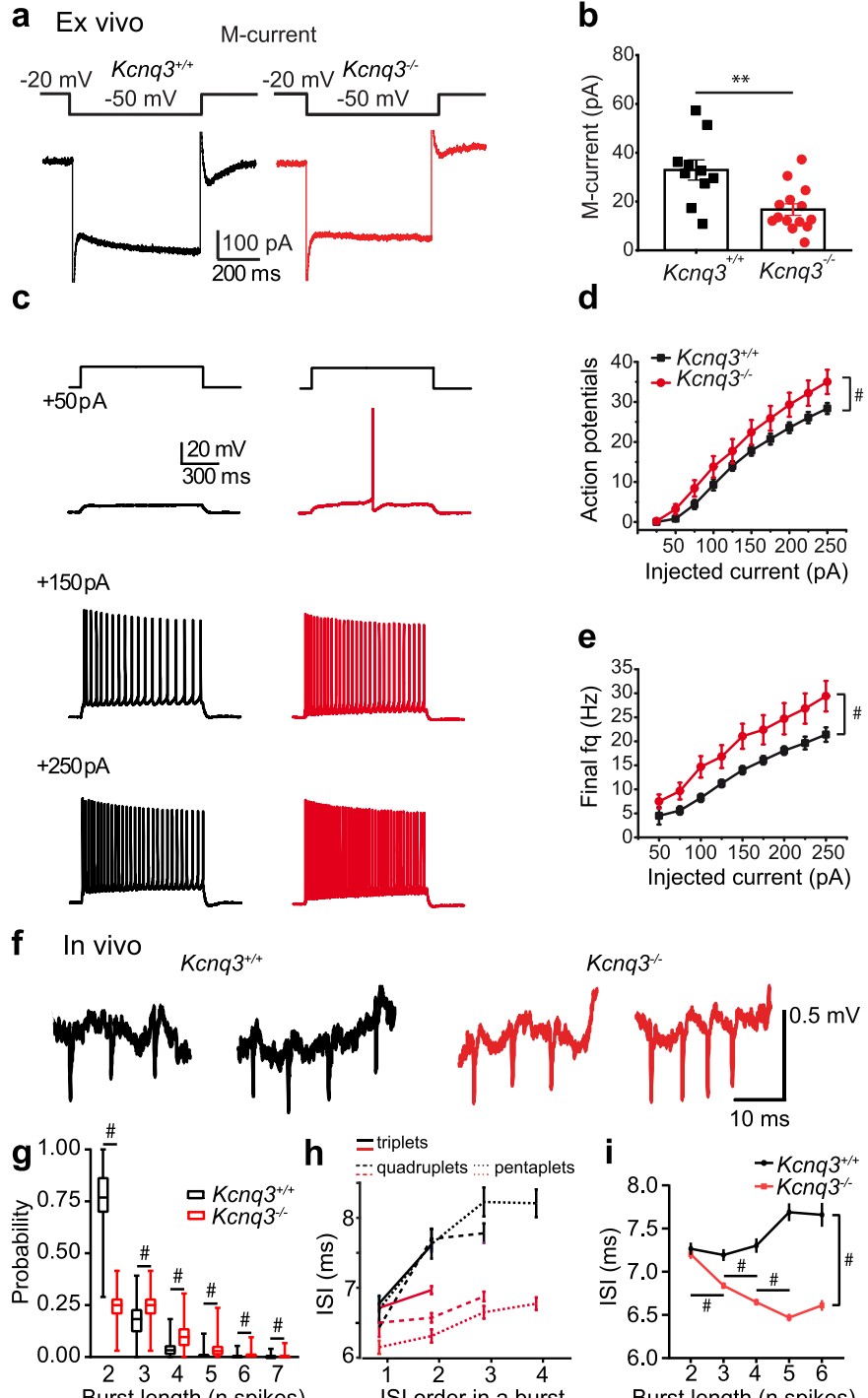

**Fig. 1 Kcnq3-containing M-channels contribute to the excitability of pyramidal cells during burst discharge. a** Representative responses of CA1 pyramidal cells in hippocampal slices in voltage clamp experiments. **b** Amplitude of M-current measured as the slow deactivating tail current ($n = 10$ cells from 3 $Kcnq3^{+/+}$ mice, black, $n = 14$ cells from 4 $Kcnq3^{-/-}$ mice, red, $p = 0.002$, t-test). The color code is used for the entire figure. **c** Representative responses of CA1 pyramidal cells in hippocampal slices to intracellular current injections of different magnitudes. **d** Number of action potentials ($n = 11$ cells from 4 $Kcnq3^{+/+}$ mice, $n = 12$ cells from 4 $Kcnq3^{-/-}$ mice, $F_{1,175} = 30.8$, #, $p < 0.0001$, ANOVA) and (**e**) their frequency (calculated for the last two action potentials, n as in (**d**), $F_{1,164} = 48.6$, # $p < 0.0001$, ANOVA) upon injection of 1 s long currents of indicated amplitudes. **f** Extracellular signal traces recorded using a silicon probe in a behaving mouse, showing representative in vivo bursts consisting of 3 and 4 spikes. **g** Histograms of relative probability of bursts with different number of spikes ($n = 299$ cells from 6 $Kcnq3^{+/+}$ mice; $n = 413$ cells from 4 $Kcnq3^{-/-}$ mice; #, $p < 0.0001$, Bonferroni tests). The center line indicates the median, the top and bottom edges indicate the 25th and 75th percentiles, respectively, and the whiskers extend to the maximum and minimum data points. **h** Spike frequency accommodation: ISI averaged according to their order in bursts containing different number of spikes (triplets, tetraplets and pentaplets). Accommodation rate was lower in $Kcnq3^{-/-}$ than in controls ($n = 305$ cells from 6 $Kcnq3^{+/+}$ mice; $n = 413$ cells from 4 $Kcnq3^{-/-}$ mice; $F_{1,44766} = 90.0$; #, $p < 0.0001$, ANOVA). **i** Average ISI for bursts of different lengths was progressively shorter with increasing burst length in $Kcnq3^{-/-}$ but not in controls (burst length x genotype: $n =$ same as in (**h**); $F_{4,7604} = 18.8$, $p < 0.0001$, ANOVA, #, $p < 0.0001$, Bonferroni tests; genotype: $F_{1,7604} = 195.2$, #, $p < 0.0001$). Data shown as mean ± SEM, see also Supplementary Information, Statistical Analysis (Fig. 1d,e, g-i).

**Table 1 Excitability of pyramidal cells.**

| | $Kcnq3^{+/+}$, $n = 11$, $N = 4$ | $Kcnq3^{-/-}$, $n = 12$, $N = 4$ |
|---|---|---|
| Resting membrane potential, mV | $-62.8 \pm 1.1$ | $-63.9 \pm 1.1$ |
| Action potential amplitude, mV | $103.3 \pm 2.6$ | $103.7 \pm 2.1$ |
| Action potential half-width, ms | $0.90 \pm 0.03$ | $0.83 \pm 0,02$ * |
| Depolarization rate during action potential, mV/ms | $362.1 \pm 19.8$ | $354.4 \pm 18.1$ |
| Repolarization rate during action potential, mV/ms | $-90.4 \pm 2.5$ | $-100.0 \pm 2.7$ * |

Data are presented as mean ± SEM, *, $p < 0.05$, $t$ test.

modulation of excitability caused by cholinergic inputs which are known to be active during theta oscillations[19,38] (Fig. 2a). We recorded the activity of 722 putative pyramidal cells (single units, Supplementary Fig. 3a) from the hippocampal CA1 area in freely behaving $Kcnq3^{-/-}$ and wild-type mice using silicon probes and computed cross-correlations of burst times in pairs of pyramidal cells during theta oscillations. In line with earlier reports[44], pyramidal cells of control mice generated bursts in a temporally coordinated fashion with an average interval between bursts of ~140 ms, i.e., at the theta band frequency of ~7 Hz (Fig. 2b). The ensemble theta-rhythmicity of bursts was abolished in $Kcnq3^{-/-}$ mice (Fig. 2b,c).

To reveal changes in the activity of individual pyramidal cells, we examined modulation of burst firing during two distinct forms of theta oscillations. We distinguished movement-related theta oscillations (type 1 theta) during spatial navigation (Fig. 2d), and immobility-related theta (Fig. 2e), which occurs in absence of movement during high alertness and sensory processing (type 2 theta[45]). The two types of theta oscillations were initially characterized based on the changes of the mesoscopic rhythm upon systemic application of muscarinic antagonists. These leave the power of local field potential (LFP) theta type 1 largely unaffected (atropine-resistant theta) while abolishing the type 2 (atropine-sensitive) rhythm[46]. Atropine sensitivity of type 2 theta is owed to its reliance on the cholinergic excitation of PV-cells in the MS[47,48], which are crucial for theta rhythm. In line with cholinergic modulation of encoding new information[38], hippocampal acetylcholine levels during behaviors associated with type 1 theta are high[49]. Behaviors connected to type 2 theta are associated with more variable levels of cholinergic stimulation[49,50] which, similar to actions of acetylcholine in the cortex, may play a role in changes of information processing during increased attention[38]. In controls, pyramidal cells generated the majority of bursts close to troughs of both movement- and immobility-related theta (Fig. 2d,e). Unlike the situation in WT mice, the population probability of burst firing in $Kcnq3^{-/-}$ mice was not entrained by movement-related theta oscillations (Fig. 2d). These changes were brought about, on the one hand, by a reduced number of cells locked to theta oscillations in the mutant (76%, control, 60%, mutant, $\chi^2(1, n = 400) = 11.2$, $p < 0.0008$, $\chi^2$ - test) and, on the other hand, by a larger spread of preferred phases of locked cells. Their phase distribution was not modulated by the theta rhythm in the mutant ($Kcnq3^{+/+}$: $n = 145$ cells, $p < 0.0001$; $Kcnq3^{-/-}$: $n = 126$ cells, $p = 0.14$, Rayleigh test, Supplementary Fig. 3b) and was better described by a mixture of two circular distributions ($Kcnq3^{+/+}$: $p = 0.374$, $Kcnq3^{-/-}$: $p = 0.032$, Silverman's bootstrap test). During immobility-related theta oscillations, mutant and control pyramidal cells fired bursts at phases that were close to oscillation troughs ($Kcnq3^{+/+}$: 344°, $n = 50$ cells, $p < 0.0001$, Rayleigh test; $Kcnq3^{-/-}$: 315°, $n = 115$ cells, $p < 0.0001$; Supplementary Fig. 3b). In agreement with the preserved phase-locking of individual cells, auto-correlations of burst times from combined theta-epochs during movement and immobility indicated a similar theta rhythmicity in individual cells of both

genotypes (Supplementary Fig. 3c). Thus, the loss of Kcnq3-containing M-channels altered periodic temporal coordination of burst discharge in CA1 pyramidal cells during spatial navigation, but not during immobility-related theta oscillations (Fig. 2f).

Dysfunction of M-type $K^+$-channels can impair coordination of burst discharge by interfering either with the generation or the readout of theta rhythm. This may involve many mechanisms, prominently including the activity of hippocampal interneurons (Fig. 2a). Average firing rates of putative fast spiking interneurons, a population controlling excitability of pyramidal cells and their synchronization during network oscillations, were unchanged in $Kcnq3^{-/-}$ mice ($Kcnq3^{-/-}$, $n = 85$ cells; $Kcnq3^{+/+}$, $n = 106$ cells, $p = 0.68$, Mann–Whitney-U-Test). The power of theta oscillations in $Kcnq3^{-/-}$ mice did not differ from controls either (Supplementary Fig. 4a). To study whether selective $Kcnq3$ ablation in pyramidal cells affects theta oscillations we generated Emx1-$\Delta Kcnq3$ mice in which the $Kcnq3$ gene was preferentially disrupted in pyramidal cells (Supplementary Fig. 4b). LFP theta oscillations were not affected in behaving Emx1-$\Delta Kcnq3$ mice (Supplementary Fig. 4c). Collectively these results suggest that the readout rather than the generation of the theta rhythm is influenced by the loss of Kcnq3 in pyramidal cells.

**Temporal disconnection of intrinsic and network-driven excitability impairs place fields generated by single spikes.** Next, we investigated how the organization of burst discharge during theta-oscillations translates into spatial representations by firing rates. Pyramidal cells fired selectively in portions of environment ("place fields") while an animal explored a rectangular arena or a circular track (Fig. 3a). Place field sizes did not differ significantly between genotypes in either enclosure (Fig. 3b, Supplementary Information, Statistical Analysis). To study spatial representations by bursts and single spikes, we computed spatial firing maps separately for spikes emitted in, or outside, of bursts. Sizes of these place fields were represented as a fraction of place fields generated by all spikes (Fig. 3a, b). Strikingly, the fields defined by the firing of single spikes were markedly smaller in $Kcnq3^{-/-}$ than in control mice (Fig. 3a, b). These changes were found both in the arena ($Kcnq3^{+/+}$: $0.46 \pm 0.03$, $n = 76$ cells from 5 mice; $Kcnq3^{-/-}$: $0.30 \pm 0.03$, $n = 95$ cells from 4 mice, $p = 0.0004$, Mann–Whitney-U-Test) and on the track ($Kcnq3^{+/+}$: $0.53 \pm 0.03$, $n = 78$ cells from 5 mice; $Kcnq3^{-/-}$: $0.40 \pm 0.02$, $n = 94$ cells from 4 mice, $p < 0.0001$, Mann–Whitney-U-Test).

How was the spatial coding by bursts influenced by the ablation of Kcnq3? The sizes of place fields of bursts were slightly larger in the mutant, but only on the circular track (Fig. 3b). On the other hand, peak as well as average rates within place fields were considerably higher in both enclosures, and the discharge during bursts was more diffusely distributed (sparse) on the track in $Kcnq3^{-/-}$ mice (Table 2). Furthermore, on the track, spatial information conveyed by spikes fired either within or outside bursts, was reduced in $Kcnq3^{-/-}$ mice (Table 2).

An earlier study suggested that a subpopulation of cells fires bursts at highest rates close to the center of place fields, where bursts overlap

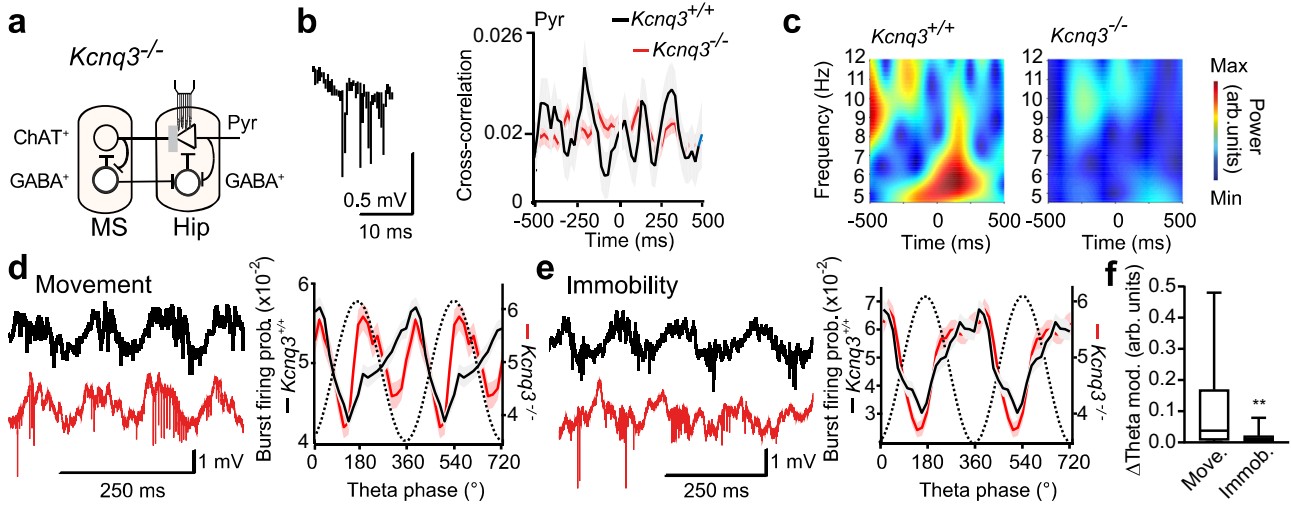

**Fig. 2 Theta-rhythmic coordination of spike bursts during spatial navigation involves Kcnq3-containing M-channels. a** Scheme: impaired modulation (gray bar) of hippocampal (Hip) pyramidal cells (triangle, Pyr) via MS cholinergic neurons (circle, ChAT$^+$) in Kcnq3$^{-/-}$ mice. Modulatory effects of cholinergic inputs are in part mediated by muscarinic receptor-mediated inhibition of Kcnq3-containing channels[19]. Circles, GABA$^+$, inhibitory cells. **b** Example complex spike burst recorded in CA1 pyramidal layer (left) and cross-correlation of burst times between pyramidal cells (right) during theta oscillations (mostly during running, mean ± SEM (shaded bonds) of 154 cell pairs with joint firing of bursts out of 625 recorded pairs (25%) from 6 Kcnq3$^{+/+}$ mice, black; 854 pairs with joint firing of bursts out of 1403 recorded pairs (61%) from 4 Kcnq3$^{-/-}$ mice, red; $\chi^2(1, n = 2028) = 227.0$, $p<0.0001$, $\chi^2$−test). The color code is used for the entire figure. **c** Spectrograms of average correlations between pairs of cells shown in (**b**), revealing theta-frequency coordination of bursts in Kcnq3$^{+/+}$ ($p = 0.007$, permutation test) but not in Kcnq3$^{-/-}$ ($p = 0.239$) mice. **d**, **e** Burst probability as a function of theta oscillation phase during spatial navigation (**d**, $n = 191$ cells from 6 Kcnq3$^{+/+}$ mice; $n = 209$ cells from 4 Kcnq3$^{-/-}$ mice) and alert immobility (**e**, Kcnq3$^{+/+}$: $n = 124$ cells; Kcnq3$^{-/-}$: $n = 210$ cells). Dotted line: reference oscillation cycle. Left: representative signals (1 Hz–10 kHz) showing theta oscillations and neuronal discharge during spatial navigation (**d**) and alert immobility (**e**) in Kcnq3$^{+/+}$ and Kcnq3$^{-/-}$ mice. As typical for the mutant, a long burst in (**d**) starts close to the theta peak. Data are presented as mean ± SEM. **f** Difference of burst probability between genotypes during navigation (**d**, top) and alert immobility (**e**, top, independent squared differences for $n = 20$ phase bins/behavior, ** $p = 0.0049$, $t$ test). The center line indicates the median, the top and bottom edges indicate the 25th and 75th percentiles, respectively, and the whiskers extend to the maximum and minimum data points. Arb. units, arbitrary units. See also Supplementary Information, Statistical Analysis (Fig. 2b–f).

with single spikes[51]. In line with those results[51], which were obtained in rats, place fields of bursts and of single spikes in Kcnq3$^{+/+}$ mice were overall correlated (z-transformed Pearson's r, arena: 0.63 ± 0.09, $n = 43$ cells; track: 0.46 ± 0.05, $n = 68$ cells). The spatial coordination was particularly high in a fraction of cells (Supplementary Fig. 5a,b). The spatial correlation of burst and single spike firing rates was markedly reduced in Kcnq3$^{-/-}$ mice compared to controls (arena: 0.42 ± 0.05, $n = 73$ cells, $p = 0.0006$; track: 0.29 ± 0.03, $n = 90$ cells, $p = 0.002$, Mann–Whitney-U-Test).

To investigate how interactions between bursts and single spikes at time scales of theta oscillations can influence spatial representations by the discharge rate we examined the influence of a burst on the probability of an ensuing single spike. In line with earlier reports[44,51], firing of a single spike increased, and later decreased again, the probability of a subsequent burst (Supplementary Fig. 5c). Furthermore, in both genotypes, the probability of single spike firing following a burst was reduced during a period of up to 50–60 ms (Fig. 3c). These time windows of lower intrinsic excitability match the ascending part of the theta cycle when firing probabilities of pyramidal cells are overall low due to the theta-rhythmic increase of inhibition. Accordingly, the recovery of intrinsic excitability after a burst coincides with times of increasing excitability during the theta cycle. The temporal match of an increased intrinsic and theta-oscillation related excitability can therefore facilitate the firing of single spikes during spatial navigation (Fig. 3c). Conversely, the timing of bursts which were no longer phase-locked during spatial

navigation due to Kcnq3 ablation (Fig. 2) disrupted the temporal match of intrinsic and theta rhythm-driven excitability. Specifically, when bursts fired more often during descending phases in Kcnq3$^{-/-}$ mice (Fig. 2d, Supplementary Fig. 3b), the firing of single spikes fell on ascending theta phases associated with high inhibition. Hence, the firing of single spikes was reduced in Kcnq3$^{-/-}$ mice (Fig. 3d) and this resulted in impaired spatial representations.

**Medial septal inputs entrain complex spike bursts.** Which neural pathways contribute to the cholinergic modulation of bursts during theta oscillations? M-type channels are inhibited by acetylcholine released from cholinergic afferents from the MS[48]. On the other hand, burst firing of pyramidal cells can be effectively controlled by perisomatic inhibition. During theta oscillations, this inhibition is rhythmically modulated by inputs from parvalbumin-positive (GABA$^+$) MS neurons[52]. To gain insights into the role of these hippocampal afferents in the theta entrainment of bursts, we recorded the discharge of putative CA1 pyramidal cells while optogenetically stimulating different inputs from MS in Kcnq3$^{+/+}$ mice. To specifically manipulate inhibitory projections from MS to the hippocampus, we introduced ChR2 in parvalbumin-positive (GABA$^+$) MS cells using a Cre-dependent virus in PV-Cre mice (Fig. 4a, b). During baseline recordings the probability of complex spike bursts changed according to the phase of theta oscillations (Fig. 4c). Optogenetic stimulation of GABA$^+$ projections phase-locked hippocampal theta oscillations

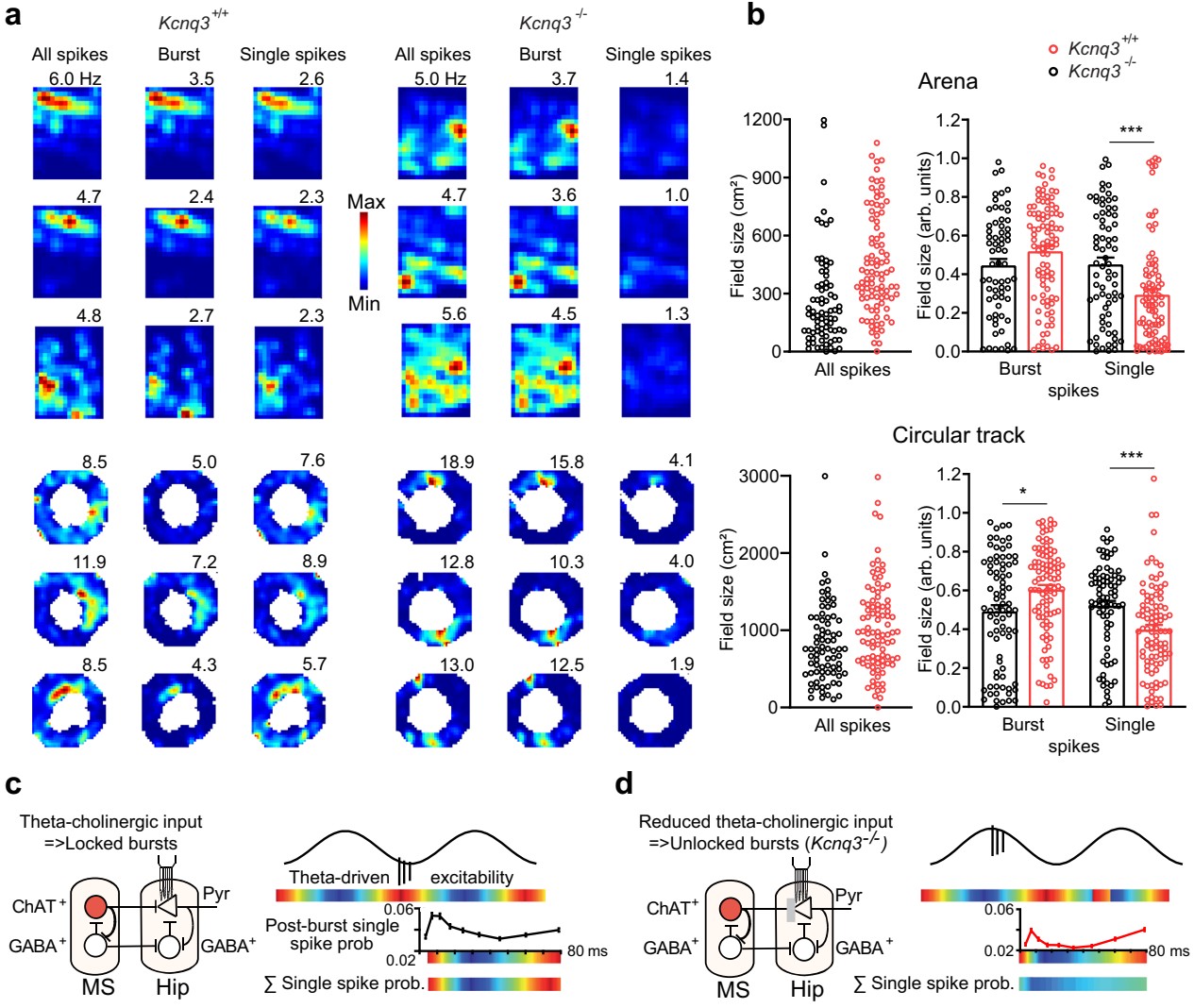

**Fig. 3 Spatial representations by bursts and single spikes in *Kcnq3⁻ᐟ⁻* and wild-type mice. a** Firing maps of CA1 pyramidal cells recorded during navigation in a rectangular arena (top) or on a circular track (bottom). For each place cell, three firing maps were constructed: using all spikes (left), spikes fired in bursts (middle) and only single spikes (right). Peak rates (Hz) indicated above of each map, color scales match across burst and single-spike firing maps in each cell. **b** Size of place fields in the arena ($n = 76$ cells from 5 *Kcnq3⁺ᐟ⁺* mice, black; $n = 95$ cells from 4 *Kcnq3⁻ᐟ⁻* mice, red, ***, $p = 0.0004$, Mann–Whitney-U-Test) and on the track (*Kcnq3⁺ᐟ⁺*: $n = 78$ cells; *Kcnq3⁻ᐟ⁻*: $n = 94$ cells *, $p = 0.013$, ****, $p < 0.0001$). The color code is used for the entire figure. Schemes illustrating the impaired spatial firing of single spikes during intact (**c**, *Kcnq3⁺ᐟ⁺*) and impaired (**d**, *Kcnq3⁻ᐟ⁻*) entrainment of burst discharge. Inset curves show probability of a single spike emission as a function of silence duration after a burst ($n = 117$ cells from 6 *Kcnq3⁺ᐟ⁺* mice; $n = 167$ cells from 4 *Kcnq3⁻ᐟ⁻* mice). Red circles, ChAT⁺ cells; GABA⁺, inhibitory cells; Pyr, pyramidal cells. The sine wave denotes theta oscillation cycle, arb. units, arbitrary units. Data are presented as mean ± SEM, see also Supplementary Information, Statistical Analysis (Fig. 3b–d).

(Fig. 4b and ref. [53]), in accord with results obtained with somatic stimulation of PV⁺ MS cells[48]. Surprisingly, the optogenetic stimulation did not entrain complex spike bursts of pyramidal cells (Fig. 4c). Hence, theta-rhythmic activity of inhibitory MS projections may not be sufficient for reliable burst entrainment.

To reveal effects of a coordinated theta-rhythmic activity of GABAergic (GABA⁺) and non-GABAergic (GABA⁻) neurons projecting to the hippocampus, we targeted in wild-type mice ChR2 to all MS cell types under the control of an ubiquitously active synthetic CAG promotor (Supplementary Fig. 6a), or to all types of MS neurons under the control of the neuron-specific human synapsin promotor (Fig. 4d). Similar to the GABA⁺ stimulation, simultaneous optogenetic stimulation of both GABA⁺ and GABA⁻ MS inputs to the hippocampus at theta frequencies entrained theta oscillations (Fig. 4e, f) and reduced running speed and its variability depending on the entrainment

fidelity (Supplementary Fig. 6b,c). This observation agrees with effects of the stimulation of GABA⁺ - projections reported earlier[53]. However, in contrast to the selective stimulation of GABA⁺ inputs, theta phase-locking of both GABA⁻ and GABA⁺ MS projections entrained burst firing of pyramidal cells similar to spontaneous theta oscillations (Fig. 4g,h, Supplementary Fig. 6d).

MS features two interconnected GABA⁻ neuronal populations projecting to the hippocampus: cholinergic (ChAT⁺) and glutamatergic cells[54]. Cholinergic cells prominently synapse on CA1 area pyramidal cells[55] and may therefore directly influence burst discharge. To investigate firing of ChAT⁺ cells in behaving mice we expressed ChR2 in the MS of *ChAT*-Cre mice using a Cre-dependent virus (Fig. 4i). MS units were recorded during spatial navigation and optogenetically identified as putative ChAT⁺ or ChAT⁻ cells based on their brief responses to series of light pulses applied at the end of each recording session

**Table 2 Properties of place cells.**

| Arena | $Kcnq3^{+/+}$, n = 79, N = 5 | | | $Kcnq3^{-/-}$, n = 96, N = 4 | | |
|---|---|---|---|---|---|---|
| | **All spikes** | **Burst** | **Single spikes** | **All spikes** | **Burst** | **Single spikes** |
| Peak rate, Hz | 4.93 ± 0.36 | 3.52 ± 0.30 | 2.60 ± 0.16 | 6.80 ± 0.46*** | 5.49 ± 0.44**** | 3.01 ± 0.20 |
| In-field rate, Hz | 2.25 ± 0.10 | 1.86 ± 0.08 | 1.58 ± 0.05 | 2.51 ± 0.09 | 2.27 ± 0.09*** | 1.69 ± 0.06 |
| Sparsity | 0.49 ± 0.03 | 0.38 ± 0.0317 | 0.54 ± 0.04 | 0.59 ± 0.04 | 0.45 ± 0.04 | 0.63 ± 0.04 |
| Inform. density, bit/spike | 1.06 ± 0.18 | 1.33 ± 0.20 | 0.82 ± 0.16 | 0.64 ± 0.09 | 1.01 ± 0.11 | 0.47 ± 0.06 |
| **Track** | $Kcnq3^{+/+}$, n = 78, N = 5 | | | $Kcnq3^{-/-}$, n = 94, N = 4 | | |
| | **All spikes** | **Burst** | **Single spikes** | **All spikes** | **Burst** | **Single spikes** |
| Peak rate, Hz | 13.29 ± 0.97 | 9.74 ± 0.82 | 6.54 ± 0.53 | 15.28 ± 0.77 | 13.41 ± 0.78*** | 5.95 ± 0.41 |
| In-field rate, Hz | 3.73 ± 0.17 | 3.20 ± 0.16 | 2.69 ± 0.19 | 4.32 ± 0.21 | 4.31 ± 0.23**** | 2.47 ± 0.09 |
| Sparsity | 0.43 ± 0.04 | 0.25 ± 0.03 | 0.44 ± 0.04 | 0.52 ± 0.04* | 0.37 ± 0.03**** | 0.50 ± 0.03 |
| Inform. density, bit/spike | 1.31 ± 0.18 | 2.18 ± 0.25 | 1.20 ± 0.17 | 0.86 ± 0.12** | 1.23 ± 0.12*** | 0.86 ± 0.13* |

Data are presented as mean ± SEM, *, $p < 0.025$ (significance level is adjusted for comparisons of the same cells in the two enclosures), **, $p < 0.01$, ***, $p < 0.001$, ****, $p < 0.0001$, Mann–Whitney-U-Test.

(Fig. 4j, Supplementary Fig. 6e). Spike waveforms of photo-responsive cells did not change during optogenetic stimulation (Fig. 4j). During movement, a fraction of MS cells displayed a clear modulation according to the phase of locally recorded LFP theta oscillation (Fig. 4k). The theta-rhythmic activity was observed in ~50% of both putative ChAT⁻ and ChAT⁺ cells (Fig. 4l).

To study the impact of cholinergic inputs from ChAT⁺ cells on theta oscillations and complex spike bursts we recorded neuronal discharge in the CA1 area during theta-rhythmic ChR2-stimulation of cholinergic projections during navigation (Fig. 4m). Somatic optogenetic stimulation of MS ChAT⁺ cells evokes theta oscillations[48,54] via intra-MS relays[48]. Accordingly the stimulation of MS-hippocampal afferents, which did not engage intra-MS projections of ChAT⁺ cells, failed to substantially affect the power of theta oscillations and did not entrain theta oscillations in a phase-synchronized fashion (Fig. 4n,o). ChAT⁺ inputs, temporally uncoupled by the stimulation from activity of GABA⁺-afferents, did not affect the probability of burst firing in pyramidal cells at time scales relevant for theta oscillations (Fig. 4p). During 30 ms light pulses none of the recorded pyramidal cells fired a number of burst spikes different from those observed during baseline epochs of the same duration ($0.051 <= p <= 0.983$, permutation test). Within 30 ms after the light pulse, two out of 18 recorded cells fired significantly more spikes within bursts ($p = 0.018, 0.044$; for non-responding 16 cells $0.082 =< p <= 0.881$, permutation test). Taken together these results suggest that the co-activation of MS GABA⁺ and ChAT⁺ inputs to the CA1 area at the time scale of the theta oscillation is necessary and sufficient for theta-rhythmic entrainment of complex spike bursts during spatial navigation.

## Discussion

In this study, we have manipulated complex spike bursts by eliminating the Kcnq3 K⁺ channel subunit and by the stimulation of main inputs from MS. We found that proper timing of complex spike bursts during theta oscillations is required for intact spatial coding by single action potentials. Kcnq3-containing channels are essential for theta entrainment of bursts during spatial navigation, but not during alert immobility. This entrainment is controlled by upstream theta-rhythmic coordination of GABAergic and cholinergic inputs from the MS to the hippocampus.

Theta-periodic burst firing by pyramidal cell populations[44] is an intriguing yet poorly understood aspect of hippocampal information processing. Theta-burst stimulation induces synaptic plasticity, suggesting that it may be relevant for the information processing involved in learning and memory. The readout of this activity can be further supported by network oscillations in downstream regions. Burst firing is sufficient for learning of hippocampus-dependent tasks[3]. Participation of neurons in temporally organized collective activity[16] depends on subthreshold resonance in individual cells, on firing rate[38], and on the modulation of membrane conductances.

Actions of acetylcholine on pyramidal cells include the inhibition of Kcnq3-containing M-channels, of slow AHP and of "leak" potassium currents[19,38,56]. Muscarinic antagonists abolish burst firing in hippocampal pyramidal cells ex vivo and in vivo[20,57]. Conversely, reduction of M-currents in mice expressing a dominant negative mutant of Kcnq2[22] results in a higher number of spikes during bursts in neonatal mice[58]. Post-synaptic M1 receptors also enhance $I_h$ and Ca²⁺-dependent cation currents while presynaptic M4 receptors modulate Kir3 K⁺-channels and voltage-dependent Ca²⁺-channels[38]. The latter effects of acetylcholine limit the spread of excitation in the CA3 region[59]. This increases the precision of spatial coding and facilitates pattern-completion during memory retrieval (reviewed in ref. [38]). High levels of acetylcholine selectively inhibit, via presynaptic M4 receptors on CA3 pyramidal cells, excitatory recurrent connections in CA3 and CA3 afferents to the CA1 area[60–62]. On the other hand, entorhinal inputs can be facilitated by acetylcholine via nicotinic receptors[63]. Modulatory actions of acetylcholine enable long-term potentiation of recently modified synapses and facilitate encoding[64,65]. The lack of theta-rhythmic offsets between bursts in pyramidal cells with impaired cholinergic modulation (Fig. 2b,c) reported here for $Kcnq3^{-/-}$ mice mimics a more correlated bursting in the absence of cholinergic stimulation during slow wave sleep[44]. Low levels of acetylcholine during slow wave sleep were suggested to result in the spread of excitatory activity and hence in the facilitation of auto-associative activity involved in retrieval[66]. While bursts are probably not directly caused by the cholinergic stimulation, but rather driven by entorhinal and CA3 inputs[67,68], the present results implicate cholinergic modulation of pyramidal cells in a proper timing of bursts within theta oscillation cycles. This adjusts population responses to afferent inputs by setting timing of bursts across

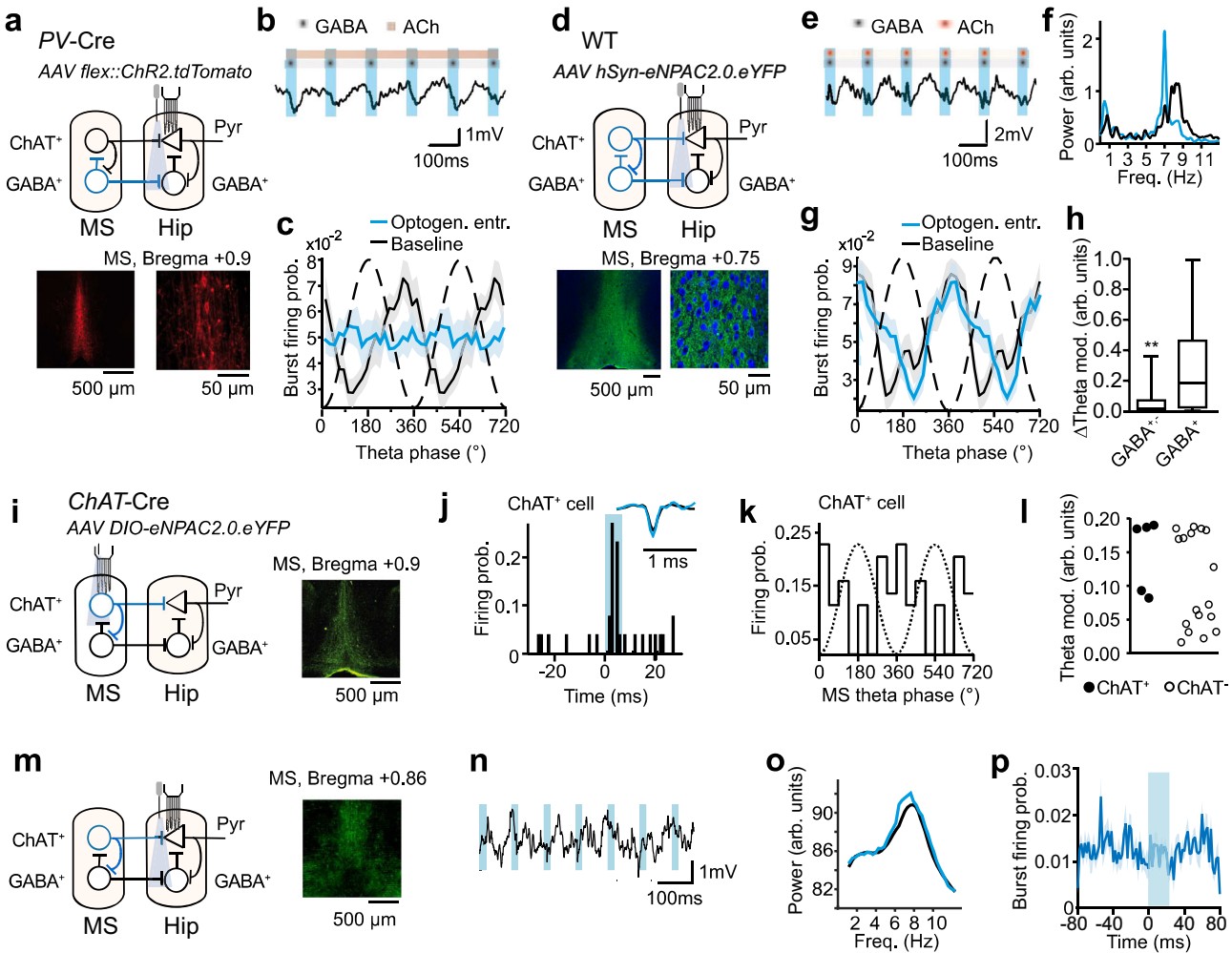

**Fig. 4 Entrainment of pyramidal cell bursts in *Kcnq3*$^{+/+}$ mice by collective rhythmicity of MS inputs. a** Scheme: optostimulation of MS GABAergic inputs (blue) to hippocampus and neuronal recordings. Images: *ChR2-tdTomato* expression. **b** Entrainment of theta oscillations in (**a**). Blue bars: light pulses; gray dots: projections of MS-GABA$^+$-neurons; red line: not stimulated cholinergic inputs. **c** Stimulation of GABA$^+$-inputs entrains oscillations but not pyramidal cells' bursts ($n = 14$ and 18 cells, 3 mice). **d** Scheme: ChR2 stimulation of MS-GABA$^+$ (gray dots) and GABA$^-$ (including cholinergic, red dots) inputs and neuronal recordings. Images: *eNPAC2.0-eYFP* expression (see also Supplementary Fig. 6a). **e** Entrainment of theta in (**d**). Dots: MS-GABA$^+$ (gray) and GABA$^-$ (red) inputs. **f** LFP power spectra during 7 Hz stimulation (blue) and baseline (black). **g** Modulation of bursts during baseline and phase-locked activation of GABA$^+$- & GABA$^-$-inputs ($n = 37$ and 38 cells, 3 mice, see also Supplementary Fig. 6d). **h** Bursts entrainment during stimulation of GABA$^+$- (**c**) vs. GABA$^+$- & GABA$^-$-projections (**d**) (independent squared differences with spontaneous theta in $n = 20$ phase bins/group, $p = 0.0055$, t test). **i** Scheme: optostimulation of cholinergic neurons and neuronal recordings in the MS. Image: *eNPAC2.0-eYFP* expression. **j** Light-triggered CCG of a putative MS-ChAT$^+$-cell (response: 5.3 SD above mean). Inset: average spikes during baseline and stimulation (Pearson's $r = 0.97$). **k** Firing probability of the cell from (**j**) vs. MS theta-phase. **l** High and low theta-rhythmic MS cells (pulled distribution's bimodality: $p = 0.005$, Silverman's test, $n = 22$ cells). ChAT$^+$-cells rhythmicity: 55th/upper 20th and upper 10th percentiles in highly and low rhythmic populations, respectively. **m** Scheme: optostimulation of cholinergic projections and hippocampal recordings. Image: *eNPAC2.0-eYFP* expression. **n** Theta oscillations are not phase-synchronized to the stimulation of cholinergic projections. **o** CA1 LFP power spectra during baseline and stimulation of cholinergic projections. **p** Stimulation of cholinergic inputs alone does not influence burst firing of pyramidal cells (spike counts, baseline vs. light, $p > 0.05$, permutation test, $n = 18$ cells, 2 mice). Data in curves are presented as mean ± SEM (shaded bonds), arb.units, arbitrary units, see also Supplementary Information, Statistical Analysis (Fig. 4h, l).

pyramidal cells and aligns the activity dynamics of individual cells with the period of the theta rhythm enabling spatial coding by single spikes.

While M/Kcnq-currents contribute to mAHP[24,69], according to previous reports[70] and the present results, their amplitudes are not affected by the lack of Kcnq3 in CA1 pyramidal cells. mAHP is therefore unlikely to mediate alterations of intrinsic bursts' features in *Kcnq3*$^{-/-}$ mice. The reduced spike frequency accommodation and the inverse coupling of burst frequency with spiking output in the mutant were not caused by changes in resting membrane potential or input resistance which, agreeing with a previous

report[26], were unchanged in the mutant. On the other hand, the observed changes in typical spiking dynamics during complex spike bursts might result from a functional interplay of Kcnq3-containing channels with other channels, e.g., large and small conductance calcium-activated K$^+$ channels[32,71]. Further investigations will be required to reveal the role of such interactions in the organization of discharge during complex spike bursts.

Cholinergic MS cells are part of a distributed theta rhythm-generating network. Similar to hippocampal interneurons, they are innervated by theta-rhythmic MS GABAergic cells[55]. Rhythmic activity of cholinergic projections from MS to hippocampus

played a central role in earlier models of the generation of theta rhythm, that postulated an important physiological role of cholinergic innervation of pyramidal cells in this process[45]. However, later studies showed that GABAergic rather than cholinergic projections are involved in theta generation[53,72]. The temporal pattern of cholinergic activity in behaving animals could not be reliably investigated before cholinergic cells in the basal forebrain could be identified with optogenetic methods[73]. Depending on behavior, acetylcholine concentration changes dynamically at the seconds-time scale in the cortex (reviewed in ref. [73]). However, these measurements currently do not permit an analysis at shorter time scales that are relevant for theta oscillations. In a recent study, optogenetic stimulation of cholinergic MS cells boosted atropine-sensitive theta oscillations indirectly via local MS connectivity. It also directly increased the precision of pyramidal cell discharge via the cholinergic MS – hippocampal pathway[48]. Here we focused on cholinergic modulation of burst discharge in behaving mice. The loss of theta entrainment of bursts in $Kcnq3^{-/-}$ mice was not due to a higher overall excitability or an increased probability of bursts in the mutant, which were rather reduced or unaffected, respectively. The observed impaired rhythmic responses of pyramidal cells might be caused, for instance, by a failure to effectively follow theta-rhythmic GABAergic inputs from MS. This scenario would be consistent with the contribution of M-currents to the subthreshold theta-band resonance in pyramidal cells[22,74]. Yet we found that intact resonant properties of pyramidal cells, in wild type mice, may not be sufficient for the entrainment of bursts by GABAergic inputs alone. A more consistent explanation for the imprecise burst discharge of $Kcnq3^{-/-}$ cells is their reduced capability to receive relevant timing signals in the absence of Kcnq3-containing M-channels. The possibility that the cholinergic inputs contribute to these entraining signals is supported, firstly, by the entrainment of bursts via the phase locked stimulation of GABAergic and non-GABAergic (mostly cholinergic) MS afferents to CA1 (Fig. 4g), secondly, by the failure of GABAergic inputs in the absence of simultaneous stimulation of non-GABAergic inputs to entrain bursts (Fig. 4c) and, thirdly, by the theta-rhythmic firing of a subpopulation of the putative ChAT+ MS cells (Fig. 4l).

During theta oscillations, somatic membrane potentials in CA1 pyramidal cells oscillate out-of-phase with the LFP rhythm in stratum pyramidale and with the membrane potential in apical dendrites[75,76]. A key mechanism of this rhythmicity is the periodic inhibition of basket interneurons by GABAergic inputs from the medial septum[72]. While the dendritic excitation that drives burst discharge[68] is maximal during phases of somatic hyperpolarization[75], bursts are typically fired during subsequent sufficiently depolarized phases which are closer to troughs. Due to their voltage dependence, M-currents are probably most active at depolarized theta phases (possibly in anti-phase with $I_h$-currents), during which bursts are mostly emitted. The present results suggest that the cholinergic inhibition of the M-current can therefore be essential for the high excitability and bursting of pyramidal cells at these phases. During these time windows, the depolarization of pyramidal cells is also mediated by the perisomatic disinhibition, timing of which is determined by GABAergic MS inputs.

Spatial navigation and alert immobility are the principal awake states associated with theta oscillations[16,45,77,78]. The release of acetylcholine in the hippocampus is higher during active than during quiet wakefulness[79]. During alert immobility, cholinergic inputs account for the sensitivity of theta oscillations to atropine, which, in turn, at least in part depends on hippocampal parvalbumin (PV)—positive interneurons[13]. The unchanged LFP theta oscillations after ablation of Kcnq3 preferentially in pyramidal cells indicate that the cholinergic input to interneurons, which are spared

by the mutation, can be essential for the theta rhythm during awake states. Apart from PV-positive basket cells, another possible target for acetylcholine are oriens-lacunosum/moleculare (O-LM) cells. Their interspike intervals are known to be regulated by Kcnq channels[23]. Overall, these results extend our view of how Kcnq3-containing M-type K+-channels regulate excitability in the hippocampus during presumably high levels of cholinergic activity. It is distinct from mechanisms involved in the generation of network oscillations, and potentially complements actions of acetylcholine during memory formation[38,79]. Elucidating a differential impact of clinically relevant $Kcnq3$-gene variants on cortical excitability and representations will likely have to rely on novel knock-in mouse models. Our work reports an important action of Kcnq3-containing M-channels in pyramidal cells. Synergistically with MS GABA-ergic inputs, these channels coordinate complex spike bursts during theta oscillations, a coordination that is required for spatial coding by single spikes. The disruption of this mechanism may be one of the functional deficits that underlie effects of cholinergic dysfunction on cognition across neuropsychiatric disorders, including, but not limited to, those associated with KCNQ3 dysfunction.

## Methods

**Animals.** Twenty-seven male mice, older than 12 weeks, were used in chronic experiments. Mice were housed under standard conditions in the animal facility and kept on a 12 h light/dark cycle, with access to food and water ad libitum. Mouse housing and experiments were in accordance with national and international guidelines and were approved by the local health authority (Landesamt für Gesundheit und Soziales, Berlin and Regierung von Unterfranken).

$Kcnq3$ constitutive knock-out ($Kcnq3^{-/-}$)[39] and $Kcnq3$ conditional knock-out mice were generated using the Cre/loxP system. The targeting vector to generate $Kcnq3^{lox/lox}$ mice was derived by cloning C57BI/6 J $Kcnq3$ genomic DNA containing exon 2–6 from a BAC-clone into the pKO901 vector containing a diphtheria toxin A cassette (DTA) for negative selection (Lexicon Genetics). A neomycin resistance ($NEO^R$) cassette flanked by FRT/loxP sites was inserted into intron 4. An additional loxP site was introduced into intron 3 for recombinant deletion of exon 4 by Cre enzyme activity. The final targeting vector containing a $Kcnq3$ DNA fragment of 4.4 kb (short arm) and 6.1 kb (long arm), a $NEO^R$ cassette flanked by FRT sites and exon 4 flanked by two loxP sites is shown in Supplementary Fig. 1. The targeting vector, which was verified by sequencing, was electroporated into R1 (129/SvJ) embryonic stem cells (ESCs). ESC clones were selected by neomycin resistance and screened for correct genomic recombination by Southern blot analysis. Selected ESCs were injected into C57BI/6 J blastocysts that were implanted into foster mothers (carried out by the Transgenic Core Facility of MDC, Berlin). Chimeric offspring was crossed with FLPe recombinase expressing mice[80] to excise the $NEO^R$ cassette. Successful recombination in F1 animals was assessed by Southern blot and PCR. Mice carrying the floxed allele ($Kcnq3^{+/lox}$) were crossed with Cre-recombinase expressing mice[81] to excise exon 4 from the $Kcnq3$ locus resulting in a frame-shift and early stop codon insertion ($Kcnq3^{+/-}$). Heterozygous mice were crossed to obtain homozygous $Kcnq3^{lox/lox}$ and $Kcnq3^{-/-}$ mice, respectively. $Kcnq3^{lox/lox}$ mice were crossed with the Emx1-ires-cre-recombinase mice[82], also in a C57BL/6 background (The Jackson Laboratory, Bar Harbor, Maine, USA), to obtain cerebral cortex-specific deletion of Kcnq3 channels. $Kcnq3^{lox/lox}$; $Emx1^{cre/+}$ and $Kcnq3^{lox/lox}$; $Emx1^{cre/cre}$ were considered conditional knock-out mice. The knock-out of $Kcnq3$ was confirmed by Southern blot, reverse transcriptase-PCR (RT-PCR, see list of primers in Supplementary Table 1) and Western blot analysis.

For optogenetic stimulation experiments $PV$-Cre knock-in mice[83] (The Jackson Laboratory), $ChAT$-Cre knock-in mice[84] (The Jackson Laboratory) and C57BL/6 male mice, 10–25 weeks old, were used. A part of the optogenetic stimulation dataset (from present $PV$-Cre mice) and of spontaneous neuronal activity recordings (in a wild-type mouse) have been used in previous reports[27,53].

**Whole cell electrophysiology.** Transverse brain (current clamp experiments) or coronal (voltage clamp experiments) slices were taken from 8 $Kcnq3$ null mice[85] and 7 wild-type controls ages P15-P20 in an ice-cold solution consisting of the following: 25 mm NaHCO$_3$, 200 mm sucrose, 10 mm glucose, 2.5 mm KCl, 1.3 mm NaH$_2$PO$_4$, 0.5 mm CaCl$_2$, and 7 mm MgCl$_2$. The cerebellum, prefrontal lobe, and temporal lobe were removed and 300 μm hippocampal slices, were cut using a Leica vibratome (Leica VT1200S). Slices were then transferred to a holding chamber containing artificial CSF (ACSF) consisting of the following: 125 mm NaCl, 26 mmNaHCO$_3$, 2.5 mm KCl, 1 mm NaH$_2$PO$_4$, 1.3 mm MgCl$_2$, 2.5 mm CaCl$_2$, and 12 mm glucose. Slices were first equilibrated at 35 °C for 30 min and maintained at room temperature (~22 °C) for ≥1 h before any electrophysiological recordings. The cutting and holding solutions (ACSF) were saturated with 95% O$_2$ and 5% CO$_2$. The experiments were

performed at 30–32 °C, with the exception of M-current measurements, which were carried out at room temperature.

Conventional whole cell patch clamp recordings were carried out on pyramidal neurons using electrodes pulled from thin-walled borosilicate glass capillaries having resistances between 2.5 and 4.5 MΩ when filled with recording solution as described below (World Precision Instruments, Sarasota, FL, USA). Pyramidal cells from the CA1 area of the hippocampus were visually identified with infrared differential interference contrast optics using a 40× water-immersion objective lens on an upright microscope (Olympus BX51, Olympus). The internal recording solution for whole-cell recording consisted of the following: 130 mM potassium methylsulfate, 10 mM KCl, 10 mM HEPES, 20 mM inositol, 4 mM NaCl, 4 mM Mg₂ATP, and 0.4 mM Na₄GTP (osmolarity, 300–305 mOsm). The pH was adjusted to 7.25–7.3 with KOH. CNQX (4 µM), APV (10 µM), and picrotoxin (100 µM) (Abcam) were present in experiments to block AMPA-mediated, NMDA-mediated, and GABA-mediated synaptic transmission. For M-current measurements 500 nM TTX, 1 mM Cs, 2 mM 4-AP, and 100 µM Cadmium we also added to the extracellular solution. All recordings were performed using a Multiclamp 700B amplifier (Molecular Devices), low-pass-filtered at 2 kHz or 10 kHz (spectral power difference, n.s., Wilcoxon signed rank test), and sampled at 10 kHz. Data were analyzed offline using Clampfit 10.7 (Molecular Devices) and MS Excel 16.

**Immunohistochemistry.** Mice were anesthetized with ketamine and xylazine and perfused through the heart with 1% (w/v) paraformaldehyde (PFA) in PBS. The brain was dissected, post-fixed for 1 h in 1% PFA at 4 °C and incubated overnight in 30% (w/v) sucrose at 4 °C and embedded in Tissue-Tek O.C.T. (Sakura). Cryosections were cut at 8 µm and blocked in 2% (w/v) BSA and 0.5% (v/v) Nonidet P-40 in PBS for 2 h. Antibodies were diluted in PBS containing 1% BSA and 0.25% Nonidet P-40 and incubated for 1 h. Nuclei were stained with DAPI, alternatively neurons were visualized with a fluorescent Nissl stain. All sections were imaged using a Zeiss LSM 510 confocal microscope. Image analysis was performed off-line with the ZEN 2009 light edition software (Zeiss) and Adobe Photoshop CS6. The following primary Kcnq3 antibody was used: rabbit anti-Kcnq3 (named Q3Crb1[28], 1:200). The secondary goat anti-rabbit Alexa 555 antibody (1:1000) was from Invitrogen.

**Virus injections.** Animals were anaesthetized with isoflurane and placed in a head frame (David Kopf Instruments, Tujunga, CA, USA). Stereotactic virus injections were performed at a rate of 100 nl min⁻¹ using a 34-gauge beveled metal needle or a glass micropipette connected via a tube to a microsyringe pump (PHD Ultra, Harvard Apparatus, Holliston, MA, USA)[53]. *PV*-Cre mice were injected in the MS (AP 0.98, L 0.0, V −5.0 and −4.5 mm) with a total of 1000 nl of Cre-dependent ChR2 (AAV2/1.CAGGS.flex.ChR2.tdTomato.WPRE.SV40, Penn Vector Core). *ChAT*-Cre mice were injected in the MS (same coordinates) with a total of 900 nl of Cre-dependent eNPAC2.0 (AAVdj-nEF-DIO-NpHR-TS-p2A-hChR2(H134R)-eYFP, a gift from Karl Deisseroth). C57BL/6 mice were injected in the MS (same coordinates) with a total of 700 nl of ChR2 (AAVdj-hSyn-NpHR-TS-p2A-hChR2(H134R)-eYFP, a gift from Karl Deisseroth, or AAV2/5.CAG.hChR2(H134R)-mCherry.WPRE.SV40, Penn Vector Core). After infusion, the injector was left at the injection site for 10 min, then slowly withdrawn and the incision was sutured.

**Implantations of optic fiber and electrodes.** Arrays of single tungsten wires (40 µm, angular cut, California Fine Wire Company, Grover Beach, CA, USA), linear silicon probes (CM32, NeuroNexus Technologies, Ann Arbor, MI, USA), octrode probes (B32, NeuroNexus Technologies) mounted on a custom-made microdrive and independently movable 8 tetrodes (fabricated from 12 µm tungsten wire, California Fine Wire Company, Grover Beach, CA, USA) loaded on a microdrive (Minidrive-8, BioSignal Group, New York, USA) were stereotactically implanted under isoflurane anesthesia either in the str. oriens, pyramidale and radiatum of the hippocampal CA1 area (AP -1.94, L 1.4, V 1.4 mm, wire arrays, CM32 probes), or in the str. oriens with subsequent positioning in the pyramidal layer (B32 probes, tetrodes), using LFP and unitary activity as a reference to record units across the pyramidal layer. A silicon probe (B32) was implanted in the MS at AP 0.98, L + 0.6, V 3.4, 8° and subsequently advanced. Reference and ground electrodes were miniature stainless-steel screws in the skull above the cerebellum. The implants were secured to the skull with dental acrylic.

Optic fibers were fabricated from 100 µm diameter fiber (0.22 numerical aperture, Thorlabs, Bergkirchen, Germany) and zirconia ferrules (Precision Fiber Products, Chula Vista, CA, USA). Optic fibers were implanted above the hippocampal pyramidal layer (AP -1.94, L 1.4, V 1.4 mm) for optogenetic entrainment of MS to hippocampus projections or in the MS (AP 0.98, L -0.6, V 3.4, 8°) for optogenetic identification of ChAT cells.

**Data acquisition and pre-processing.** Electrodes were connected to operational amplifiers (Neuralynx, Bozeman, MT, USA or Noted B.T., Pecs, Hungary) to eliminate cable movement artifacts. Electrophysiological signals were differentially amplified, band-pass filtered (1 Hz–10 kHz, Digital Lynx, Neuralynx) and acquired continuously at 32 kHz. Recordings were performed as animals explored a gray wooden rectangular arena (50×30×18 cm), a circular track (diameter 60 cm) or

rested in home cage. A light emitting diode was attached to the headset to track the animal's position at 25 frames/s.

**Optogenetic stimulation.** Implanted optic fibers were connected to a 473-nm diode-pumped solid-state laser (Laserglow Technologies, Toronto, ON, Canada) via a fiberoptic patch cord (Thorlabs) and a zirconia sleeve (Precision Fiber Products). The laser was controlled by a stimulus generator (Multichannel Systems, Reutlingen, Germany). For the stimulation of the MS to hippocampus pathway 30 ms pulses of blue light (473 nm) were applied in the hippocampus at frequencies of 6, 7, 8, 9, 10, or 12 Hz. For optogenetic stimulation of MS ChAT cells 5 ms pulses of blue light were applied in the MS at frequencies of 2 or 5 Hz. Light power at the tip of the patch cord during light on-phase of pulses patterned according to the above protocols was 5–10 mW (measured with PM100D power meter, Thorlabs).

**Histology.** After completion of the experiments, mice were deeply anaesthetized and electrolytic lesions at selected recording sites were performed. Then the animals were perfused intracardially with saline followed by 4% paraformaldehyde in PBS and decapitated. Brains were fixed overnight in 4% paraformaldehyde, equilibrated in 1% PBS for an additional night, sectioned in 40 µm slices using an oscillating tissue slicer (EMS 4500, Electron Microscopy Science, Hatfield, PA, USA) and stained with cresyl violet to confirm recording sites.

**Data analysis.** Electrophysiological data were preprocessed using NeuroSuite (https://sourceforge.net/projects/neurosuite) and analyzed using custom-written MATLAB algorithms (MathWorks, Natick, MA, USA). LFP was obtained off-line by down-sampling of the wide-band signal to 1250 Hz using Neurophysiological Data Manager. Action potentials were detected in a high-pass filtered signal. Spike waveforms were extracted and represented by the first three principal components and by amplitudes of action potentials. Spike sorting was performed automatically using KlustaKwik followed by manual clusters adjustment using Klusters[86]. Putative pyramidal cells and interneurons were identified based on their auto-correlograms and firing rate (<3 Hz for pyramidal cells, >7 Hz for putative basket cells). Single pyramidal cells with a clear refractory period (<3 ms) as well as single and multiunit interneurons were used in further analysis. Isolation distance was computed for sorted units (putative pyramidal cells, $Kcnq3^{+/+}$: 48.9 ± 1.2, $n = 309$; $Kcnq3^{-/-}$: 67.2 ± 2.1, $n = 413$). Bursts were defined similar to earlier reports[51] as a series of spikes separated by intervals of no more than 15 ms. Time stamp of the first spike in a burst was considered as a time stamp of the burst.

Firing maps of pyramidal cells were computed by dividing the number of spikes in a given spatial pixel (2×2 cm) by the time spent in this pixel. Periods of immobility were excluded from the analysis. Peak firing rate was defined as the maximum firing rate over all pixels in the environment. Place fields were detected as spatially continuous areas where the firing rate exceeded 1 Hz. Sparsity, a measure of firing field compactness was computed as in[13].

Theta oscillations were detected automatically in the recordings during running based on theta/delta power ratio. Theta oscillation phase was extracted by the Hilbert transform of 5–10 Hz filtered signal[53]. Phase histograms of individual spike trains were computed and normalized, first, by the deviation (if any) of the underlying phase distribution from uniformity in respective phase bins and, second, by the total number of events. Circular uniformity (Rayleigh) test, mean phase and the resultant vector length were estimated for each histogram. Histograms of preferred discharge phases were computed by grouping mean phases of individual cells. Individual unit discharge probability histograms were convolved with the Gaussian kernel of size 2 SD. Burst probability – theta phase histograms during immobility and movement were computed for theta episodes during running (>3 cm/s) and immobility (<2 cm/s). Continuous Morlet wavelet transform (<40 Hz) was used to detect theta-oscillatory components in unitary discharge cross-correlograms. For spectral analysis of co-firing, the probability at zero time was computed as the most likely −200 to 200 ms average lag between bursts in pairs of spike trains. Power spectral density (PSD) of LFP was computed using the multitaper method (NW = 3, window size 1024).

In optogenetic stimulation experiments, for each stimulation epoch entrainment fidelity was computed within 10 sec intervals as the ratio of cumulative PSD around the optogenetic stimulation frequency (± 0.5 Hz) to the cumulative PSD in the theta band[53]. Stimulation epochs with mean theta-entrainment fidelity > 0.3 were defined as epochs with high entrainment. Bursts were detected during theta episodes in baseline (when stimulation was not applied) and during stimulation epochs with high entrainment.

For the identification of ChaT⁺-cells stimulation with 2 Hz and 5 Hz (epoch duration is 4 min each, pulse duration is 5 ms) was applied in the *ChAT*-Cre mouse transduced with a Cre-dependent opsin construct in the MS. Cells with a clear refractory period (<3 ms) were used in further analysis. To estimate responses of the cells to the stimulation, a cross-correlation with the times of light stimulation on the interval [−10 ms, 10 ms] around the light stimulus was computed for each cell (size of the bin is 1 ms). Maximum number of spikes within the bin before ($A_{obs}$) and after ($B_{obs}$) the light stimulus was identified for real data. Then the train of light stimuli was shifted to the baseline for a random time interval from the beginning of stimulation $N = 10000$ times. Maximum number of spikes within the bin before and after the light stimulus were identified for each shift $i$ and surrogate distributions $A_i$ and

$B_i$, $i = 1..N$ were obtained. For each cell p-values were defined as $p_1 = P(A_i > A_{obs})$ and $p_2 = P(B_i > B_{obs})$. A cell was classified as a putative ChaT$^+$- cell if at least for one stimulation epoch it was light-responsive after light stimulus ($p_2 < 0.05$) and non-light-responsive before the pulse ($p_1 \geq 0.05$).

**Statistics and reproducibility.** The statistical significance of single comparisons was determined by the Mann-Whitney-U-test or with t-test. The normality of distributions was determined by the Lilliefors test. P-values <0.05 were considered to indicate significance, statistical tests were two-sided except for the one-sided Silverman's bootstrap test of bimodality in Fig. 4l. The center line in box and whiskers plots indicates the median, the top and bottom edges indicate the 25th and 75th percentiles, respectively, and the whiskers extend to the maximum and minimum data points. In analyses that involved two or more factors, analysis of variance (ANOVA) was applied. For representative images and recordings, similar data were collected from at least 3 different animals. A detailed description of statistical comparisons is provided in Supplementary Information, Statistical Analysis.

**Reporting summary.** Further information on research design is available in the Nature Research Reporting Summary linked to this article.

## Data availability
Source data underlying Figs. 1–4, Tables 1,2 and Supplementary Figs. 1–6 are provided with the paper (Source Data file). Spike trains recorded in $Kcnq3^{-/-}$ and $Kcnq3^{+/+}$ mice were made available via Figshare (https://doi.org/10.6084/m9.figshare.14785434). Further datasets generated during the current study are available from corresponding authors upon reasonable request.

## Code availability
All the codes used in the current study are available from corresponding authors upon reasonable request.

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

## Acknowledgements
We would like to thank Dr. R. Sahdev and Dr. M. Larkum for providing *ChAT*-Cre mice, Dr. K. Deisseroth for providing AAVdj-eNPAC and M. Zeller for the valuable input during preparation of the manuscript.

## Author contributions
X.G., F.B., C.C., M.C.-C. M.A. and T.K. performed in vivo electrophysiological experiments, M.H. developed genetic mouse models, M.H. and S.S. generated genetically modified mice, S.S. performed IH, X.G., M.A., M.G., F.B., M.C.-C., M.-A.C. and A.P. analysed in vivo electrophysiological data; H.S. performed ex vivo electrophysiological experiments, A.V.T. designed and supervised ex vivo electrophysiological experiments, A.P. and T.K. supervised in vivo electrophysiological part, T.J.J. supervised molecular biological and cellular imaging experiments, A.P. and T.J.J. wrote the paper with input from all authors, all authors contributed to the study design, T.J.J. initiated the study.

## Funding
This work was supported by a SAW grant of the Leibniz Gemeinschaft (TJJ) and Deutsche Forschungsgemeinschaft (DFG; Exc 257 NeuroCure, TJJ, TK and AP; SPP1665, 1799/1-2, Heisenberg Programme, 1799/3-1, AP). Open Access funding enabled and organized by Projekt DEAL.

## Competing interests
The authors declare no competing interests.
