## [Peer Review File · Nature Communications]

REVIEWER COMMENTS

Reviewer #1 (Remarks to the Author):

The study of Gao et al "Place fields of single spikes in hippocampus involve KCNQ3 channel-dependent entrainment of complex spike bursts" aims to investigate regulation of neuronal communication in the hippocampus and medial septum by testing the model of a balanced spatial code by complex spike bursts mediated by neuromodulatory and inhibitory inputs during theta oscillations. Notwithstanding quite interesting data collected and wide variety of the employed methods, the study contains numerous major and minor issues based on which the significant revision of the manuscript is suggested, additional analysis of the induced action potential spike activity are required, plus some additional experiments should be performed.

Major comments

Interestingly, whether the resting membrane potential in Kcnq3^{-/-} or Kcnq3^{+/+} mice neurons were different or not? What are these resting membrane potential values in Kcnq3^{-/-} or Kcnq3^{+/+} mice neurons? Whether Kcnq3^{-/-} neurons displayed higher frequency of spontaneous action potential compare to Kcnq3^{+/+} neurons?

Also, the authors recommended to perform voltage-clamp step protocol recording to assess and compare transmembrane current amplitudes and activation-inactivation window current for voltage-gated sodium channels in the neurons from both Kcnq3^{-/-} and Kcnq3^{+/+} groups.

- Figure 2, b, page 6 line 176

(Kcnq3^{-/-}: n=1307 cell pairs, Kcnq3^{+/+}: n=48 cell pairs) – why such a huge x27 times difference in the tested cell numbers between Kcnq3^{-/-} or Kcnq3^{+/+} mice neurons?

- Line 180-182

"d, Kcnq3^{-/-}, n=113 cells; Kcnq3^{+/+}, n=22 cells) and alert immobility (e, Kcnq3^{-/-}, n=122 cells; Kcnq3^{+/+}, n=24 cells)" – why the number of Kcnq3^{-/-} neurons x5 times higher vs. Kcnq3^{+/+} neurons? It is hardly possible that the authors put together such unequal number of experiments to reduce SEM error bars in one group to achieve statistically significant difference between the groups, however such striking difference should be sorted with additional experiments for the wild type group. On the contrast, the authors had close values of the number of the cells for Figure 1 b, n=12 and n=11 for Kcnq3^{-/-} or Kcnq3^{+/+} groups. To rectify this negative aspect of the study the authors have to increase n- numbers for the cells of Kcnq3^{+/+} group and point out the number of the cell isolations for the corresponding series of experiments (N). To be fair Supplemental Information in Statistical Analysis contains means, SEM and n values but not for all datasets. N- numbers are missing throughout the whole study.

What was the number of the animals from which the cells were isolated in both Kcnq3^{-/-} or Kcnq3^{+/+} mice groups? Please provide these data.

- Figures 2, 3 and 4 have no numerical characterisation of the illustrated data in either text or figure legend. There are only n-numbers that are still unequal for Kcnq3^{-/-} or Kcnq3^{+/+} groups.

- Same inequality is in the Figure 3 legend, lines 253-254

"b Size of place fields in the arena (Kcnq3^{-/-}, n=70 cells; Kcnq3^{+/+}, n=20 cells) and on the track (Kcnq3^{-/-}, n=58 cells; Kcnq3^{+/+}, n=11 cells)". And in lines 258-259: "(Kcnq3^{+/+}, n=63 cells, Kcnq3^{-/-}, n=153 cells)"

- Same inequality is in Figure 4 legend panel f line 323

"...burst probability during spontaneous theta and theta-entraining stimulation (n=43 and 7 cells, respectively).

- Discussion, lines 383-384

The authors refer to the medium after-hyperpolarization (mAHP) assessed in a different study, however by some reason they did not provide this information from their own experiments, which could be easily calculated from the induced action potential data (Figure 1a). The authors also ignored very informative parameters such as: threshold, depolarisation and repolarisation rates

and half width of the induced action potential.

- Table 1. Properties of place cells, line 663-664 contains means +/- SEM values but no n numbers which is rather confusing but very typical for this manuscript. Again, the authors should add both numbers: n (number of individual cells) and N (number of animals/isolations).

Minor comments

- Abstract page 2 line 42

"KCNQ3-containing M-type K⁺ channels" – since KCNQ is a gene it must be italic, however within the context it is rather a protein subunit which KCNQ encodes. Therefore, it is more correct to write "Kv7.3-containing M-type K⁺ channels".

- Abstract page 2 line 49-51

"Our results suggest that imbalanced representations of spatial location by bursts and single spikes may underlie cognitive disabilities associated with KCNQ3-mutations". – this is an interesting hypothesis, however it would be nice if the authors provided an experimental evidence for the statement running experiment with KCNQ3-mutations, which are associated with epilepsy. By the way, was there any correlation found between KCNQ3-derived mutation epilepsy and cognition and/or orientation pathotypes? Also, here KCNQ gene is written in capital letters whereas in the most places in the text *Kcnq*, is there any reason for that? If not please make uniform.

- Introduction page 3 line 76

"KCNQ2/KCNQ3 and KCNQ5/KCNQ3 voltage-gated potassium channels" – change for Kv7.2/7.3 and Kv7.5/7.3, respectively, because it talks about the subunits, not alleles.

- Introduction page 3 line 76

"Jointly with KCNQ2, KCNQ3 subunits" – Change for Kv7.2 and Kv7.3, respectively.

Everywhere in the text when it is about subunits, not alleles, please change KCNQ for Kv7 to avoid confusion.

- Introduction page 3 line 76

"Mutations in the genes encoding either subunit of heteromeric KCNQ2/3 voltage-gated potassium channels have been linked to childhood epilepsy³⁰⁻³²" – Great, is there any direct cross-link between KCNQ3-induced epilepsy and cognition and/or orientation was reported?

- Results page 4, lines 122-123

"... average firing rates in the mutant (*Kcnq3*^{-/-}, 1.08±0.03 Hz, *Kcnq3*^{+/+}, 1.45±0.07 Hz, *p*<0.0001, Mann-Whitney-U-Test)" – no n-numbers are provided in the text, please add.

- Results page 4, lines 128-129

"(interspike intervals, *Kcnq3*^{-/-}, 6.66±0.02 ms, *Kcnq3*^{+/+}, 7.90±0.09 ms, *F*_{1,5615}=289, *p*<0.0001, ANOVA)" – same, no n-numbers are provided in the text, please add. There are numbers in the figure legend (*n*=12 and *n*=11, respectively), but that will be so much better if these numbers are also in the Results text.

- Results page 4, lines 133,

"... intraburst frequency (i.e. shorter ISIs)" – i.e. as well as other Latin in the text must be italic.

- page 5 line 165

~140 ms, i.e. – i.e. as well as other Latin in the text must be italic.

- Figure legend 2, line 185, panel f

"... during navigation vs. alert immobility" – vs. as well as other Latin in the text must be italic.

- Figure legend 4, line 326, panel g

"... GABA⁺ vs." – vs. as well as other Latin in the text must be italic.

- Results page 7, lines 195-196,

"medial septum (MS)" – no need to depict MS again as it has been done in the text above.

- Results page 10, lines 271,

"...of different sets of inputs from the medial septum (MS)" – no need to depict MS again as it has been done in the text above.

- Figure legend 1, line 144, panel d

"d. Signal traces showing representative bursts in vivo..." – What is the nature of these "Signal traces", one might think that this is either extracellular or current-clamp whole-cell recordings, unfortunately it is not explained either in the Results text or Figure 1 legend, which is really confusing.

- Discussion lines 360-361

"... coordination of inhibitory and cholinergic MS" – it does not sound good, should be either "inhibitory and excitatory" or "GABAergic and cholinergic"

- Methods, Whole cell electrophysiology, line 619 and 620

The authors sampled the changes of membrane potential at 10 kHz and filtered at 2 kHz, whereas normally the frequency of sampling should be roughly twice as much of the filtration frequency, whereas here the difference is 5-fold. This unnecessary increases the size of the *.abf files and the recorded signal becomes rather noisy, so for the further analysis should be filtered offline, which again increases the weight of each trace.

- In the Results, line 112-116, the authors speculate that "Kcnq3^{-/-} mice likely form increased levels of KCNQ2 homomeric instead of KCNQ2/3 heteromeric channels, resulting in a large decrease of M-current magnitude since currents through heteromeric KCNQ2/3 channels are much larger than those mediated by homomeric KCNQ2 channels." The authors did not explain why knocking out of KCNQ3 might affect expression of other KCNQ genes. They also do not mention that the current amplitude through the homomeric Kv7.2 channel is lower than that for the heteromeric Kv7.2/7.3 due to significantly (~30-fold) lower sensitivity of Kv7.2 to Pi(4,5)P2 bisphosphate compare to that for Kv7.3.

Dr Vsevolod (Seva) Telezhkin (PhD, FHEA)
Lecturer in Physiology
School of Dental Sciences
Faculty of Medical Sciences
Framlington Place
Newcastle University
Newcastle upon Tyne NE2 4BW
Tel: +44 (0) 191 20 88240

Reviewer #2 (Remarks to the Author):

This article presents data on the effect of constitutive removal of the gene *Kcnq3* coding a protein subunit KCNQ3 of the M current (as well as testing selective removal of KCNQ3 in pyramidal cells). The authors focus on analyzing the effect of this removal on hippocampal pyramidal cells. They show in slice preparations that knockout of this M current subunit results in pyramidal cells that respond to current injection with a larger number of action potentials and a higher final firing rate. They show with in vivo recordings that *Kcnq3^{-/-}* animals show longer spiking bursts with shorter interspike intervals and less spike frequency accommodation in bursts fired by hippocampal neurons. They show a striking loss of theta rhythmic cross-correlations between bursts in the *Kcnq3^{-/-}* animals, and a reduction in the theta phase specificity of bursts. They also show a striking reduction in *Kcnq3^{-/-}* in the size of place cell firing fields that involve single spikes (potentially due to the longer bursts in *Kcnq3^{-/-}* animals). Finally, they show results of rhythmic activation of GABAergic or GABA and non-GABAergic cells on the timing of bursts in hippocampus. Overall, these are interesting and clinically relevant effects of KCNQ3 knockout on the detailed dynamics of hippocampal neurons that help understand the functional role of cholinergic modulation and address the effect of a gene shown to be important in epilepsy and developmental disorders. However, the clarity of the presentation needs to be improved to make these results accessible to the reader and explain the relevance of some sections of Figure 4.

Major comments:

1. The description of some of the major findings is unclear in the abstract, even for a person knowledgeable about this field. The results in the figures were clearer and they need to make a better effort to summarize these results in the abstract, text and figure legends. The main problems with the abstract are that they are trying to provide too much interpretation of mechanism instead of just stating results.

1A. For example, the phrase: "facilitated high-frequency discharge of individual pyramidal cells disrupted" – This sounds like it is an experimental manipulation involving direct facilitation, but it is instead referring to the hypothesis that the indirect effect of *Kcnq3*^{-/-} knockout on high frequency discharge is causing this effect. They should instead cut most of this phrase and just indicate that lack of *KCNQ3* resulted in less theta rhythmic bursting.

1B. Similarly, "impaired a balanced contribution of these firing modes" is very unclear. They should simply state the clear end result that lack of *KCNQ3* resulted in reduction in size of place fields defined by single spikes.

2. Page 3 – "model of a balanced spatial code" – They don't really present a model in this paper, so they should remove the word "model" and phrase this differently, focusing on their experimental data indicating an interaction of coding by bursts and single spikes. The abstract and this section presents the work in a rather diffuse and theoretical framework that haven't been fully proven. They would need more experiments and network modeling to fully prove these very broad statement about "the balance of bursting and single spikes mediated by neuromodulation and inhibitory inputs." They should instead remove these overly broad statements and focus on clearly presenting their important and significant experimental results.

3. There are many results presented in the figures that do not seem to have supportive statistical tests in the figure legends or text. For example the results shown in Figures 1b, 1c do not seem to have statistical tests presented in the text or figure legend or supplemental materials (they also need to make much clearer which statistical results are presented only in the supplemental materials).

4. When the statistics for figures are provided in the supplemental figures, this needs to be stated more clearly in the main figure legend, with an explicit statement of WHICH parts of the figure has the statistics shown in the supplemental section. Otherwise, the reader misses these statistics as noted in a few points in this review. The reader should not be expected to guess about which statistics are in the supplemental section.

5. Figure 2b,c – They show a striking effect on theta rhythmicity in the cross-correlation of bursts. But this raises the important question of the effect on autocorrelation of bursting activity. They need to show autocorrelations as well.

6. Figure 2d legend and figure – "representative hippocampal LFP signal" They seem to only show theta rhythmic LFP for the wild type? They need to show examples of theta rhythmicity for both *Kcnq3*^{+/+} and *Kcnq3*^{-/-} and provide some summary statistics (power spectra) even if they are indicating a negative result on LFP.

7. Figure 3b – The effect on single spike place fields is striking and it is nice to see it replicated in both arena and track. This interesting result needs to be stated much more clearly and simply in the abstract and main text rather than be obscured by vague pronouncements about the "balance" of bursts and single spikes.

8. Line 213- "LFP theta oscillations were not affected (sup Fig 2c) – In the supplemental materials, they show the power spectra for the theta LFP from the pyramidal selective mutant (*Emx1-Kcnq3*^{-/-}) but they need to show the power spectra comparison and example theta LFP from the constitutive mutant as well (as noted above).

9. Line 352 – "temporally coordinated activity" – This conclusion of the Results section is very unclear. Overall, the significance of the optogenetic manipulations of the medial septum in Figure 4 are not made sufficiently clear and the rationale and significance of these results relative to

Kcnq3^{-/-} should be made clearer in the Results and Discussion section.

Specific comments:

Page 2 – “Firing of place cells... is driven by signals of self-motion and spatial cues...” They should not state this hypothesis as proven fact. Could add “appear to be” before driven.

Page 2 – Supported by experience-dependent inputs... Again they are presenting hypotheses from previous work as proven fact. Should tone this down.

Page 3 – Jointly with KCNQ2, KCNQ3... Regulate the availability... This is a very dense sentence. They should expand this sentence into a few sentences to indicate that KCNQ3 is expressed in both axons and somatodendritic areas and to describe the potential differential roles.

Page 3 – “dampening inhibition” – this is unclear and should be expanded if there is space

Line 122 – “without increased average firing rates of the mutant” – They seem to be presenting numbers for the significant effect on bursts, but they should also present the numbers for the lack of significant effect on average firing rates – and be clear about which data address each point.

Line 132-133 – This is a very confusing sentence that tries to merge two results about number of spikes and intraburst frequency. They should split this into a 2-3 sentences describing the results separately.

Page 4 and Figure 1b,c legend (n=12 cells, n=11 cells). They do not seem to provide any statistical test for this difference in number and frequency of action potentials in slice preparations. They need to provide the statistical results for 1b and for 1c in the figure legend and/or the text.

Figure 1g – They show statistical significance within Kcnq3^{-/-} but they should present the comparison of the Kcnq3^{+/+} and ^{-/-} (black versus red). In general, they need to present their statistical results more clearly in all of their figure legends or text, as some statistical comparisons shown in the figures do not seem to be addressed in the text or figure legends.

Figure 2 legend – “impaired readout (grey bar) of inputs from MS cholinergic neurons” – The term “readout” does not seem at all appropriate for describing the modulatory activity of cholinergic neurons. Cholinergic modulation is instead probably modulating the input-output dynamics of hippocampal circuits. “readout” should be changed to “modulatory effects” or something like that.

Line 176 - Page 6 – “48 pairs” Why are there so few Kcnq3^{+/+} pairs?

Line 178 – What is “Gaussian surprise”?

Line 181 - Figure 2e – How can the Kcnq3^{-/-} be entrained to theta but not show theta rhythmic cross-correlations? This is confusing?

Line 194 – “consistently high or more variable levels of acetylcholine” – This section is confusing and should be split into more sentences that describe their point in more detail. In particular, they need to make clear that previous work showed that type 1 theta does not depend upon Ach and that type 2 does depend upon Ach and cite Kramis and Vanderwolf.

Line 200 - Figure 2d – “mutant was not modulated” – This is confusing as the figure in 2d shows clear rhythmicity of the bursting relative to theta at double the frequency. They need to mention this and provide statistical tests showing lack of modulation to support this statement.

Line 203 Figure 2f – What is being shown in Figure 2f? Is it the difference between ^{+/+} and ^{-/-}? If so, this needs to be much clearer as the figure legend makes it sound as if Figure 2f is only showing results from Kcnq3^{-/-} and does not mention that it is a difference between ^{-/-} and ^{+/+}.

Line 224 – “display smaller place fields” – Again there needs to be some presentation of statistical

results to support this statement. The lack of statistical tests is surprising throughout the manuscript.

Line 254 – “n=11 cells” – why are there so few Kcnq3+/+ cells?

Line 257 – “inset curves” – Are the insets on the same scale? What is the scale? Why is there a large increase at later times in the red plot for -/- This is surprising.

Throughout the figures they use the abbreviation “(au)” without definition. Presumably au means arbitrary units, but they should define this in the legends.

Line 263 – “which were no more phase-locked” – do they mean “no longer phase locked”?

Line 278 – citation 49 – should this also include citation 46?

Line 293 – “the former synapse” – This is confusing as they are citing the paper by Robinson on glutamatergic inputs. Perhaps that paper did present results about the cholinergic input but maybe the phrasing of the sentence would be clearer without using the word “former”

Line 308 – figure 4a – “blue contours” – where are the blue contours in Figure 4a? Do they mean in Figure 4b?

Line 313 – red stripe – what are the blue bands in Figure 4b?

Lines 338-339 and discussion – I believe that tests of entrainment of theta by cholinergic versus GABAergic modulation of hippocampus were done in the Dannenberg paper 46. Probably what is different here is the focus on entrainment of bursts, but they need to be much clearer about this.

Line 373 – “as well as the mediated via” – This is an awkward sentence structure and needs to be rewritten.

Line 379 – The link and citation regarding cholinergic modulation is appropriate but could be made clearer with more detail and more citations. (Hasselmo, 2006 provides a concise overview).

Line 422 – unchanged LFP theta oscillations after ablation – This did not seem to be shown explicitly in the figures or with statistical tests anywhere in the figures, the text or the supplemental materials. This comparison needs to be provided as both a figure and text.

Line 512 – “a circular track” – what is the symbol? Is that meant to be diameter? Should just use the word diameter

Reviewer #3 (Remarks to the Author):

GENERAL COMMENT

The authors study how KCNQ3 channels affect spike bursts, single spikes, theta entrainment, and place fields in CA1 hippocampus. They employ an impressive variety of techniques – constitutive and conditional knockout mice, in vitro and in vivo electrophysiology, optogenetic stimulation and identification, and behavior – and the result is a set of interesting and potentially important observations. What is lacking, though, is a consistent through-line: a connecting theme to link the different parts of this paper together.

As it stands, the paper has two major parts: (1) a study of KCNQ3 knockout physiology (Figures 1-3) and (2) a study of how the different types of inputs from medial septum (MS) affect theta entrainment in CA1 (Figure 4). These two parts are not unrelated – M-type channels are blocked by acetylcholine and MS provides acetylcholine to CA1 – but the connection seems weak. It is not clear how the results of part 1 should affect the reader’s understanding of part 2 (and vice versa).

A related opinion: it is also not clear (to this reader, at least), how the major observation of Figure 2 (that theta coordination of spike bursts is disrupted after KCNQ3 knockout in moving mice) explains the major observation of Figure 3 (that single-spike place field sizes are smaller in the knockouts). Maybe the data of Figure 3c,d explain this, but if so, more explication would be appreciated.

SPECIFIC COMMENTS

(A) In vitro electrophysiology. The in vivo data of Figures 1-3 depend entirely on the full KCNQ3 knockout (the pyramidal cell-specific version is used only to make a small point). Given its importance, the authors should characterize CA1 pyramidal neuron intrinsic properties in this knockout mouse more thoroughly than they do. At present, the characterization is limited to injecting a family of current steps and measuring f-I curves. That is a good start – and the fact that they found a difference between control and knockout in this way is remarkable since an earlier study (Ref. 27) did not – but it is only a start.

Some suggestions:

A1. Resting potential, resting input resistance, impedance. The authors suggest in the Discussion (lines 387-390) that the principal effect of eliminating KCNQ3 on intrinsic excitability is mediated by resting potential rather than medium afterhyperpolarization (mAHP). This might be true, but resting potential is not reported in this manuscript, nor is resting input resistance, and mAHP is not measured. While theta is not generated locally in CA1, entrainment to theta might be affected by the intrinsic resonance properties of CA1 neurons (see, e.g., Hu, Vervaeke, and Storm 2002), which might in turn depend on the M current. This issue might be probed, as Hu and colleagues do, by using a ZAP current (sinusoid of increasing frequency) to measure impedance.

A2. M current. The KCNQ3 $-/-$ mouse does not lack M current in CA1 neurons because it continues to express KCNQ2 (Supplementary Figure 1). As the authors note (lines 112-116) this “likely” means that M currents are reduced in the knockout. It would be much more useful if they could show what the differences are, by directly measuring whole cell M currents in voltage clamp.

A3. If the authors do more in vitro experiments, they really should do so in older animals. The animals used here were so young (P15-20) that, not only is there a mismatch with the animals used in the in vivo experiments (12 weeks), but the “other ionic mechanisms” that the authors postulate (lines 390-393) affect spike discharge are still developing.

(B) Lines 120-123. The authors note that average firing rates in vivo are not increased in the knockout. In fact, they are significantly decreased. Why is this point passed over without comment?

(C) Movement vs immobility. The authors demonstrate in Figure 2d,e that theta coordination of spike bursts is abnormal in the knockout case when the animals are moving but not when they are immobile. This is a very interesting result, but I question this line: “In contrast, the population probability of burst firing in the mutant was not modulated by movement-related theta oscillations (Fig. 2d).” (Lines 199-201). To my eye, Figure 2d shows a frequency of twice theta: one peak on the decaying phase and a second peak on the rising phase. These were population data (113 cells combined into a mean and SEM). It would be useful for the reader to know more about what the population distribution was. From Figure 2 alone, one could imagine a scenario where there were two ensembles of bursting cells that were theta-locked but asynchronous with each other. This scenario is unlikely, but I use it simply to argue that a fuller accounting of the population is in order.

(D) The spike burst abnormality of Figure 2 is not carried over into Figure 3, where only single spikes show a place field abnormality. As I noted above, I do not understand the relationship between the Figure 2 and Figure 3 results – or, more generally, what the authors mean when they write, as in the Abstract, about a “balanced contribution of these firing modes in place fields.” What does balance mean exactly?

(E) The demonstration that, in mice, both GABAergic and cholinergic inputs from MS to CA1 are required for spike burst entrainment (Figure 4) is good. What would be better is if the authors could explain or even just speculate (in Discussion) why this is. Also, as noted above, it would be exceptionally good – for the sake of a through-line – if they could say quite what the results of Figure 4 have to do with the M current, which after all is the main subject of the paper.

Detailed response to the reviewers (Gao et al.)

We thank all three reviewers for the time they have taken to critically evaluate our manuscript and for their detailed and insightful comments. Following their advice, we have performed a substantial number of new experiments, improved the clarity of presentation, and include a careful, detailed statistical analysis of our results. We believe that our revised manuscript is now significantly improved.

Reviewer #1 (Remarks to the Author):

The study of Gao et al "Place fields of single spikes in hippocampus involve KCNQ3 channel-dependent entrainment of complex spike bursts" aims to investigate regulation of neuronal communication in the hippocampus and medial septum by testing the model of a balanced spatial code by complex spike bursts mediated by neuromodulatory and inhibitory inputs during theta oscillations. Notwithstanding quite interesting data collected and wide variety of the employed methods,

We thank the reviewer for these positive comments about our data and methods.

the study contains numerous major and minor issues based on which the significant revision of the manuscript is suggested, additional analysis of the induced action potential spike activity are required, plus some additional experiments should be performed.

Major comments

Interestingly, whether the resting membrane potential in Kcnq3^{-/-} or Kcnq3^{+/+} mice neurons were different or not? What are these resting membrane potential values in Kcnq3^{-/-} or Kcnq3^{+/+} mice neurons? Whether Kcnq3^{-/-} neurons displayed higher frequency of spontaneous action potential compare to Kcnq3^{+/+} neurons?

This is indeed an important question which we have now addressed in detail. The resting membrane potential did not differ between genotypes (Kcnq3^{+/+}, -62.8 ± 1.1 mV, Kcnq3^{-/-}, -63.9 ± 1.1 mV, $p = 0.4$, t-test). These new results are presented in lines 119-122 and in the new Table 1. Since the frequency of spontaneous action potentials of CA1 pyramidal cells is rather low and variable in brain slices, we report the frequency of evoked firing (Fig 1). We also provide a detailed account of in vivo spontaneous firing rates for all spikes, and separately for single spikes and spikes within bursts on a large sample of neurons in lines 132-139.

Also, the authors recommended to perform voltage-clamp step protocol recording to assess and compare transmembrane current amplitudes and activation-inactivation window current for voltage-gated sodium channels in the neurons from both Kcnq3^{-/-} and Kcnq3^{+/+} groups.

We carefully considered this experiment and identified the following concerns that limit its implementation and interpretation. Space clamp in slices is poor, resulting in low accuracy of sodium current recordings. Moreover, work from Kole and colleagues has shown that the midpoint of inactivation of axonal and somatic sodium current differs by 20mV (Battfeld et al., 2014). Considering that Kcnq3-containing channels are

mostly axonal, measurements of sodium currents at somata will not accurately reflect relevant effects of Kcnq3 deletion on activation/inactivation of sodium channels. The measurements of action potential amplitudes and depolarization (lines 125-126 and Table 1) are consistent with similar properties of voltage-gated sodium channels in Kcnq3^{-/-} and Kcnq3^{+/+}.

Figure 2, b, page 6 line 176 (Kcnq3^{-/-}: n=1307 cell pairs, Kcnq3^{+/+}: n=48 cell pairs); why such a huge x27 times difference in the tested cell numbers between Kcnq3^{-/-} or Kcnq3^{+/+} mice neurons?

This may indeed seem strange, but we had not measured more control cells since the theta rhythmic pattern of co-firing (in a sound number of measured control cells) was similar to published data (Mizuseki et al., 2012). By additional recordings during the revision, and by analysis of a previously recorded Kcnq3^{+/+} dataset, we now increased the number of controls and provide a more balanced number of pairs for the genotypes (154 cell pairs with joint firing of bursts out of 625 recorded pairs (25%) from 6 Kcnq3^{+/+} mice; 854 pairs with joint firing of bursts out of 1403 recorded pairs (61%) from 4 Kcnq3^{-/-} mice; $\chi^2(1, n = 2028) = 227.0, p < 0.0001, \chi^2$ - test, lines 197-199).

To further rule out that the lack of theta-coordination in the mutant was due to a larger sample we randomly subsampled it to the size of the control distribution (n = 154). The low coordination of bursts in the mutant reported in the Figure 2 was highly representative of the theta coordination in random subsets of Kcnq3^{-/-} cell pairs (Supplementary Statistics, lines 142-146).

Line 180-182

Kcnq3^{-/-}, n=113 cells; Kcnq3^{+/+}, n=22 cells) and alert immobility (e, Kcnq3^{-/-}, n=122 cells; Kcnq3^{+/+}, n=24 cells) why the number of Kcnq3^{-/-} neurons x5 times higher vs. Kcnq3^{+/+} neurons? It is hardly possible that the authors put together such unequal number of experiments to reduce SEM error bars in one group to achieve statistically significant difference between the groups, however such striking difference should be sorted with additional experiments for the wild type group. On the contrast, the authors had close values of the number of the cells for Figure 1 b, n=12 and n=11 for Kcnq3^{-/-} or Kcnq3^{+/+} groups. To rectify this negative aspect of the study the authors have to increase n- numbers for the cells of Kcnq3^{+/+} group and point out the number of the cell isolations for the corresponding series of experiments (N). To be fair Supplemental Information in Statistical Analysis contains means, SEM and n values but not for all datasets. N-numbers are missing throughout the whole study.

Of course, we did not use different sample sizes to manipulate statistical significance. The rather low yet sound number of cells presented in the previous submission reliably showed the entrainment of bursts in wild types. However, we agree with the reviewer that this skewed number of measured cells may raise doubts and have therefore performed more experiments. We increased the number of cells by recording 242 new single units and by adding further control datasets recorded in mice of the same wild-type genetic background. The number of cells is now similar for Kcnq3^{+/+} and Kcnq3^{-/-} mice. Additionally, we now provide the number of animals for each experiment in Figure legends and/or in the Results text instead of pointing to the number of mice in each experimental preparation in methods.

What was the number of the animals from which the cells were isolated in both Kcnq3^{-/-} or Kcnq3^{+/+} mice groups? Please provide these data.

The cells were recorded in 4 Kcnq3^{-/-} and 6 Kcnq3^{+/+} mice, with recordings of place cells in 5 Kcnq3^{+/+} mice. These numbers are now provided for each experiment.

Figures 2, 3 and 4 have no numerical characterisation of the illustrated data in either text or figure legend. There are only n-numbers that are still unequal for Kcnq3^{-/-} or Kcnq3^{+/+} groups.

We thank the reviewer for pointing out that the reliance on graphical presentation of statistics and reference to the Supplement for more detailed information was not optimal. We have now substantially extended the numerical presentation of results in the main text pertinent to Figures 2, 3 and 4 and provide p-values of theta entrainment statistics, performed in response to reviewers' requests (lines 234, 237-239, 241-242), a detailed numerical account of single spike place fields (lines 274-277), of spatial correlations between burst and single spike firing fields (lines 288-293). We also included in figure legends exact p-values (for $p \geq 0.0001$) for the graphically presented statistical results and increased measurements to obtain more balanced sample numbers. All results reported in the paper are now accompanied by numerical information about sample sizes and statistical significance.

Same inequality is in the Figure 3 legend, lines 253-254. Size of place fields in the arena (Kcnq3^{-/-}, n=70 cells; Kcnq3^{+/+}, n=20 cells) and on the track (Kcnq3^{-/-}, n=58 cells; Kcnq3^{+/+}, n=11 cells). And in lines 258-259:(Kcnq3^{+/+}, n=63 cells, Kcnq3^{-/-}, n=153 cells).

After additional recordings of place cells, we extended sample sizes for experiments in arena/track are: n = 76/78 cells from 5 Kcnq3^{+/+} mice; n = 95/94 cells from 4 Kcnq3^{-/-} mice (lines 274-277) and in the Figure 3cd, Kcnq3^{+/+}, n = 117 cells from 6 Kcnq3^{+/+} mice, Kcnq3^{-/-}, n = 167 cells from 4 Kcnq3^{-/-} mice (lines 323-324).

Same inequality is in Figure 4 legend panel f line 323 burst probability during spontaneous theta and theta-entraining stimulation (n=43 and 7 cells, respectively).

Indeed, there was an imbalance in sample sizes, which we eliminated by new experiments during the revision. Since the viral vector used in the original dataset was not available during the revision, we now used an equivalent AAV vector driving ChR2 specifically in neurons under the control of synapsin promotor in further 3 mice and recorded 37 and 38 pyramidal cells (single units) during spontaneous theta and theta-entraining stimulation, respectively. The entrainment of bursts using ChR2 expression in all types of MS neurons is now shown in Figure 4g. The new results are in full agreement with our earlier findings that were based on ChR2 expression in all MS cells. This included glial cells which, however, are not stimulated by light delivered on hippocampal projections of MS neurons (now shown as a Suppl. Figure 6a,d).

Discussion, lines 383-384

The authors refer to the medium after-hyperpolarization (mAHP) assessed in a different study, however by some reason they did not provide this information from their own experiments, which could be easily calculated from the induced action

potential data (Figure 1a). The authors also ignored very informative parameters such as: threshold, depolarisation and repolarisation rates and half width of the induced action potential.

*We appreciate this useful suggestion. To address this request, we have analyzed these aspects in earlier recorded data and found that mAHP amplitude before and after apamin application, action potential amplitude and the maximal depolarization rate did not differ between genotypes. Threshold of action potential could not be reliably estimated with the used recording parameters. However, the maximal repolarization rate was faster and half width of the induced action potential was reduced in the mutant compared to the control. This result and the reduction of the M-current in *Kcnq3*^{-/-} mice agrees with the report of Simkin et al. (2021) who showed that similar changes of excitability accompany a prolonged reduction of the M-current in iPSCs. Thus, their study together with our new measurements support the point made earlier in the Discussion, i.e. that interactions of *Kcnq3*-containing channels with other potassium conductances (e.g. calcium-activated K⁺ channels) may underlie changes of the intrinsic organization of burst discharge in the mutant. The new results are presented in Table 1 and in Suppl. Figure 2.*

Table 1. Properties of place cells, line 663-664 contains means +/- SEM values but no n numbers which is rather confusing but very typical for this manuscript. Again, the authors should add both numbers: n (number of individual cells) and N (number of animals/isolations).

We are sorry about this confusing presentation. The number of cells in the Table was the same as in Figure 3b and therefore was not reiterated. Now n (number of cells) and N (number of animals) is stated as requested by the reviewer throughout the manuscript, including Table 2.

Minor comments

Abstract page 2 line 42

“KCNQ3-containing M-type K⁺ channels”: since KCNQ is a gene it must be italic, however within the context it is rather a protein subunit which KCNQ encodes. Therefore, it is more correct to write Kv7.3-containing M-type K⁺ channels.

*As mentioned by the reviewer, italic KCNQ3 and *kcnq3* refer to genes in humans and mice, respectively, and non-italic KCNQ3 and *Kcnq3* to respective proteins / channels. We have referred to Kv7.2, Kv7.3, Kv7.5 when introducing these channels, but prefer to use the older *Kcnq* instead of the newer Kv7 nomenclature to indicate the channels – to be consistent with our previous work, but also to prevent confusion because *Kcnq3*^{-/-} is used consistently and correctly throughout the manuscript.*

Abstract page 2 line 49-51

“Our results suggest that imbalanced representations of spatial location by bursts and single spikes may underlie cognitive disabilities associated with KCNQ3-mutations.” This is an interesting hypothesis, however it would be nice if the authors provided an experimental evidence for the statement running experiment with KCNQ3-mutations, which are associated with epilepsy. By the way, was there any correlation found between KCNQ3-derived mutation epilepsy and cognition and/or orientation

pathotypes? Also, here KCNQ gene is written in capital letters whereas in the most places in the text Kcnq, is there any reason for that? If not please make uniform.

We thank the reviewer for raising this point. In the Abstract and Introduction we mention that mutations of the KCNQ3-gene and its expression were not only linked to epilepsy (Miceli et al., 2015), but also to types of cognitive impairment that include autism spectrum disorders and intellectual disability (Gilling et al., 2013, Kaminsky et al., 2015, Lauritano et al., 2019, Herrero et al., 2020). While patients with benign familial neonatal convulsions in general lack marked cognitive impairments, a fraction of them displays recurrent seizures and develop severe cognitive deficits. Studies of KCNQ3-gene variations go beyond the scope of the present work, yet we agree that these data will be important and now discuss it (lines 554-555). We also further clarify that the human KCNQ3-gene is referred to in the abstract (line 50) and state that impaired hippocampal representations, which in particular in humans go beyond the spatial domain, may contribute to the spectrum of cognitive deficits associated with KCNQ3 mutations (lines 48-50). According to established nomenclature, human genes and proteins are written in capital letter e.g. KCNQ3, while for mouse it is lower case (Kcnq3), with italics indicating that the gene is meant.

Introduction page 3 line 76

“KCNQ2/KCNQ3 and KCNQ5/KCNQ3 voltage-gated potassium channels”: change for Kv7.2/7.3 and Kv7.5/7.3, respectively, because it talks about the subunits, not alleles.

Introduction page 3 line 76

Jointly with KCNQ2, KCNQ3 subunits; Change for Kv7.2 and Kv7.3, respectively.

Everywhere in the text when it is about subunits, not alleles, please change KCNQ for Kv7 to avoid confusion.

Please see explanation provided above for the use of the nomenclature used in this study. While using mainly the KCNQ nomenclature, we have also explained in line 75 the correspondence to the Kv7 nomenclature that was later proposed for the proteins/channels only.

Introduction page 3 line 76

“Mutations in the genes encoding either subunit of heteromeric KCNQ2/3 voltage-gated potassium channels have been linked to childhood epilepsy”; Great, is there any direct cross-link between KCNQ3-induced epilepsy and cognition and/or orientation was reported?

We have specified now that cognitive impairments we refer to were observed also in patients with epilepsy, linked to KCNQ3-mutations (Miceli et al., 2015, Lauritano et al., 2019, lines 88-90).

Results page 4, lines 122-123

“average firing rates in the mutant (Kcnq3^{-/-}, 1.08 +/- 0.03 Hz, Kcnq3^{+/+}, 1.45 +/- 0.07 Hz, p<0.0001, Mann-Whitney-U-Test”); no n-numbers are provided in the text, please add.

Please now find n in the analysis of firing rates in lines 132-139.

Results page 4, lines 128-129

(interspike intervals, Kcnq3^{-/-}, 6.66 ± 0.02 ms, Kcnq3^{+/+}, 7.90 ± 0.09 ms, F_{1,5615}=289, p<0.0001, ANOVA; same, no n-numbers are provided in the text, please add. There are numbers in the figure legend (n=12 and n=11, respectively), but that will be so much better if these numbers are also in the Results text.

Following the reviewer's suggestion, we now made sure that n is provided or reiterated for each result in the Figure legend or in the Results text, next to each result (for related results, n is usually provided in the beginning of the paragraph). On some occasions, to make the text easier readable, we provide this information in Figure legends, which can be easily accessed by the reference to Figures cited in Results, with more extended statistical analysis corresponding to several figure panels given in the Supplement.

Results page 4, lines 133,

intra-burst frequency (i.e. shorter ISIs; i.e. as well as other Latin in the text must be italic.

page 5 line 165 “~140 ms, i.e.” i.e. as well as other Latin in the text must be italic.

Figure legend 2, line 185, panel f

“during navigation vs. alert immobility” vs. as well as other Latin in the text must be italic.

Figure legend 4, line 326, panel g

GABA⁺ vs.; vs. as well as other Latin in the text must be italic.

We are not completely sure whether “i.e.” and other common Latin should be written as italic and leave the choice of the style for later editing by the Journal.

Results page 7, lines 195-196,

medial septum (MS; no need to depict MS again as it has been done in the text above.

Thanks, corrected.

Results page 10, lines 271, of different sets of inputs from the medial septum (MS) no need to depict MS again as it has been done in the text above.

Has been modified accordingly.

Figure legend 1, line 144, panel d

Signal traces showing representative bursts in vivo. What is the nature of these “Signal traces”, one might think that this is either extracellular or current-clamp whole-cell recordings, unfortunately it is not explained either in the Results text or Figure 1 legend, which is really confusing.

Thank you for pinpointing this issue. We apologize for not mentioning this important detail in a figure in which several methods are combined. As now specified in the Figure legend, the extracellular signals were recorded by a silicon probe (lines 153-

154). We have now left in only high-resolution burst traces (Figure 1f) that are directly related to the analysis shown in this figure.

Discussion lines 360-361

coordination of inhibitory and cholinergic

it does not sound good, should be either “inhibitory and excitatory” or “GABAergic and cholinergic”

You are right, we have changed to GABAergic and cholinergic (lines 446-447).

Methods, Whole cell electrophysiology, line 619 and 620

The authors sampled the changes of membrane potential at 10 kHz and filtered at 2 kHz, whereas normally the frequency of sampling should be roughly twice as much of the filtration frequency, whereas here the difference is 5-fold. This unnecessary increases the size of the *abf files and the recorded signal becomes rather noisy, so for the further analysis should be filtered offline, which again increases the weight of each trace.

That is indeed a reasonable alternative for the presented analysis of action potentials frequency. However, sampling at a higher rate made it possible to assess now a number of action potential properties with an acceptable resolution (for this analysis) of 100 microseconds. No changes have been made.

In the Results, line 112-116, the authors speculate that "Kcnq3^{-/-} mice likely form increased levels of KCNQ2 homomeric instead of KCNQ2/3 heteromeric channels, resulting in a large decrease of M-current magnitude since currents through heteromeric KCNQ2/3 channels are much larger than those mediated by homomeric KCNQ2 channels." The authors did not explain why knocking out of KCNQ3 might affect expression of other KCNQ genes. They also do not mention that the current amplitude through the homomeric Kv7.2 channel is lower than that for the heteromeric Kv7.2/7.3 due to significantly (~30-fold) lower sensitivity of Kv7.2 to Pi(4,5)P2 bisphosphate compare to that for Kv7.3.

We appreciate the comment. However, the KO of Kcnq3 did not change the expression levels of Kcnq2, as shown in Suppl. Fig. 1c. We write “Disruption of Kcnq3 ... did not lead to a compensatory upregulation of Kcnq2 expression” (line 110-112 and refer to our result in Supplement (Suppl. Fig. 1c)).

In the absence of any upregulation, with equal transcription of Kcnq2 in the presence or absence of Kcnq3, you will (1) eliminate the efficient Kcnq2/Kcnq3 channels (of course). (2), since Kcnq2 now lacks Kcnq3 as binding partner, this will increase the number of homomeric Kcnq2 channels (which, depending on relative abundancies of both subunits and protein-protein interaction affinities, may also form to a minor degree when both subunits are expressed). This is stated as ‘Hence, Kcnq3^{-/-} mice likely form increased levels of Kcnq2 homomeric at the expense of Kcnq2/3 heteromeric channels. ...’ in lines 113-114. We now also point to differences in sensitivity to Pi(4,5)P2 bisphosphate as a possible direct mechanism for the reduced responsiveness to the excitability modulation via M-receptors in Kcnq3^{-/-} mice, thank you for pointing this out (lines 117-119).

Reviewer #2 (Remarks to the Author):

This article presents data on the effect of constitutive removal of the gene *Kcnq3* coding a protein subunit KCNQ3 of the M current (as well as testing selective removal of KCNQ3 in pyramidal cells). The authors focus on analyzing the effect of this removal on hippocampal pyramidal cells. They show in slice preparations that knockout of this M current subunit results in pyramidal cells that respond to current injection with a larger number of action potentials and a higher final firing rate. They show with *in vivo* recordings that *Kcnq3*^{-/-} animals show longer spiking bursts with shorter interspike intervals and less spike frequency accommodation in bursts fired by hippocampal neurons. They show a striking loss of theta rhythmic cross-correlations between bursts in the *Kcnq3*^{-/-} animals, and a reduction in the theta phase specificity of bursts. They also show a striking reduction in *Kcnq3*^{-/-} in the size of place cell firing fields that involve single spikes (potentially due to the longer bursts in *Kcnq3*^{-/-} animals). Finally, they show results of rhythmic activation of GABAergic or GABA and non-GABAergic cells on the timing of bursts in hippocampus. Overall, these are interesting and clinically relevant effects of KCNQ3 knockout on the detailed dynamics of hippocampal neurons that help understand the functional role of cholinergic modulation and address the effect of a gene shown to be important in epilepsy and developmental disorders.

We thank the reviewer for appreciating the relevance of our findings.

However, the clarity of the presentation needs to be improved to make these results accessible to the reader and explain the relevance of some sections of Figure 4.

Major comments:

1. The description of some of the major findings is unclear in the abstract, even for a person knowledgeable about this field. The results in the figures were clearer and they need to make a better effort to summarize these results in the abstract, text and figure legends. The main problems with the abstract are that they are trying to provide too much interpretation of mechanism instead of just stating results.

We are sorry about the lack of clarity of results description in the abstract. As suggested by the reviewer, interpretations are now left out and main results are stated. We also make a connection to the optogenetic experiments, as also suggested by reviewer 3, and feel that the abstract has improved considerably.

1A. For example, the phrase: “facilitated high-frequency discharge of individual pyramidal cells disrupted” This sounds like it is an experimental manipulation involving direct facilitation, but it is instead referring to the hypothesis that the indirect effect of *Kcnq3*^{-/-} knockout on high frequency discharge is causing this effect. They should instead cut most of this phrase and just indicate that lack of KCNQ3 resulted in less theta rhythmic bursting.

*Thank you. This sentence has been modified as suggested: “In mice lacking functional *Kcnq3*-containing M-type K⁺ channels, we found that pyramidal cell bursts are less coordinated by the theta rhythm than in controls during spatial navigation, but not during alert immobility.” (lines 41-43).*

1B. Similarly, “impaired a balanced contribution of these firing modes” is very unclear. They should simply state the clear end result that lack of KCNQ3 resulted in reduction in size of place fields defined by single spikes.

Indeed, this statement was confusing, we now state in the abstract the consistent reduction of single spikes place fields in two- and one-dimensional environments: “Place fields of single spikes recorded in one- and two-dimensional environments were smaller in the mutant.” (lines 45-46)

2. Page 3 model of a balanced spatial code; They don’t really present a model in this paper, so they should remove the word and phrase this differently, focusing on their experimental data indicating an interaction of coding by bursts and single spikes. The abstract and this section presents the work in a rather diffuse and theoretical framework that haven’t been fully proven. They would need more experiments and network modeling to fully prove these very broad statement about the balance of bursting and single spikes mediated by neuromodulation and inhibitory inputs. They should instead remove these overly broad statements and focus on clearly presenting their important and significant experimental results.

Following the suggestion of the reviewer, we have removed references to balanced spatial representations and modeling – indeed, they were not the focus of the present experimental work. We also reformulated the last paragraph of the Introduction to highlight the main findings of this study (lines 93-105).

3. There are many results presented in the figures that do not seem to have supportive statistical tests in the figure legends or text. For example the results shown in Figures 1b, 1c do not seem to have statistical tests presented in the text or figure legend or supplemental materials (they also need to make much clearer which statistical results are presented only in the supplemental materials).

We apologize for not providing these essential details in a more accessible way. This is now corrected throughout, including for the experiments shown in Figures 1b, 1c, for which we performed additional statistical comparisons between the genotypes (current Figures 1d, 1e, lines 149-153). See also our response to reviewer 1.

4. When the statistics for figures are provided in the supplemental figures, this needs to be stated more clearly in the main figure legend, with an explicit statement of WHICH parts of the figure has the statistics shown in the supplemental section. Otherwise, the reader misses these statistics as noted in a few points in this review. The reader should not be expected to guess about which statistics are in the supplemental section.

We thank the reviewer for this suggestion. At the end of each figure legend, we now specify those panels for which statistical results are described in the Supplementary Information.

5. Figure 2b,c They show a striking effect on theta rhythmicity in the cross-correlation of bursts. But this raises the important question of the effect on autocorrelation of bursting activity. They need to show autocorrelations as well.

This comment is well taken. Newly computed autocorrelations of bursts times during theta epochs had a characteristic appearance with a leading peak in the theta band in both genotypes (Suppl. Figure 3c). The rhythmicity of autocorrelations in the mutant did not differ from controls (cumulative probability of the discharge with theta-lags, (Supplement, lines 39-40) Thus, pyramidal cells in the mutant are on average entrained (according to a sensitive circular statistics on individual cells, 16% less of entrained cells during running in the mutant than in controls, lines 232-235), yet during running at more variable preferred phases than in controls (as shown now in more detail in Supplementary Figure 3b). This phase variability also agrees with a less coordinated theta coordination across pyramidal cells (Figure 2 b, c). Our analysis of autocorrelations (shown in Suppl. Figure 3c) is now described in Results (lines 242-245). We feel that this is an important addition to our paper.

6. Figure 2d legend and figure, representative hippocampal LFP signal. They seem to only show theta rhythmic LFP for the wild type? They need to show examples of theta rhythmicity for both *Kcnq3*^{+/+} and *Kcnq3*^{-/-} and provide some summary statistics (power spectra) even if they are indicating a negative result on LFP.

*We now provide examples of theta rhythmic signals in Fig 2d and 2e for both genotypes and show power spectra of LFP during theta epochs as a Suppl. Figure 4a. The cumulative theta band power was not different between *Kcnq3*^{+/+} and *Kcnq3*^{-/-} (Results, lines 254-255, Suppl. Figure 4a, Supplement, lines 44-46).*

7. Figure 3b The effect on single spike place fields is striking and it is nice to see it replicated in both arena and track. This interesting result needs to be stated much more clearly and simply in the abstract and main text rather than be obscured by vague pronouncements about the balance of bursts and single spikes.

We thank the reviewer for appreciating this finding, which was reproduced and reinforced by the extended dataset acquired during the revision. This important result is now more clearly stated in the abstract, introduction and results. The difference of sizes of single spike and burst place fields became even more evident when analysed as a fraction in place fields composed of all spikes. These results are reported in Fig 3 b and described in lines 45-46, 100-102, 272-277, and Suppl. Information., lines 152-159.

8. Line 213 LFP theta oscillations were not affected (sup Fig 2c). In the supplemental materials, they show the power spectra for the theta LFP from the pyramidal selective mutant (*Emx1-Kcnq3*^{-/-}) but they need to show the power spectra comparison and example theta LFP from the constitutive mutant as well (as noted above).

We agree and now provide a quantification of the statistical comparison of theta power in the legend of Supplemental Figure 3c (Supplement, lines 48-49) and show example theta LFP (and neuronal discharge) both in the constitutive mutant and control (Figure 2 d, e).

9. Line 352; temporally coordinated activity; This conclusion of the Results section is very unclear. Overall, the significance of the optogenetic manipulations of the medial septum in Figure 4 are not made sufficiently clear and the rationale and significance

of these results relative to *Kcnq3*^{-/-} should be made clearer in the Results and Discussion section.

*We agree that the connection of the optogenetic experiments to the other results concerned mainly with effects of *Kcnq3* may not have been immediately evident. We have reformulated the last sentence of the Results (lines 435-438), improved the introduction of optogenetic studies in the Results (lines 332-336) and extended the joint discussion of genetic ablation and optogenetic experiments, indicating the rationale for particular experiments and stating the significance and connections between their results (lines 510-539). In the Abstract, we now introduce this connection by writing ‘Less modulated bursts, followed by a post-burst pause of single spike firing, offset network oscillatory and intrinsic excitability. ... Optogenetic manipulations of upstream signals revealed that neither medial septal GABA-ergic nor cholinergic inputs alone, but rather their joint activity, is required for entrainment of bursts.’ (lines 43-48).*

Specific comments:

Page 2; Firing of place cells; is driven by signals of self-motion and spatial cues; They should not state this hypothesis as proven fact. Could add “appear to be” before driven.

Corrected accordingly: “Firing of place cells in the dorsal CA1 area⁶ appears to be driven by signals of self-motion and spatial cues from entorhinal cortex grid and border cells, respectively⁷⁻⁹.” (lines 65-66).

Page 2; Supported by experience-dependent inputs; Again they are presenting hypotheses from previous work as proven fact. Should tone this down.

We have reformulated: “The firing of upstream CA3 place cells is more experience-dependent and has been shown to encode substantially different environments¹⁰.” (lines 67-68).

Page 3; Jointly with *KCNQ2*, *KCNQ3* regulate the availability; This is a very dense sentence. They should expand this sentence into a few sentences to indicate that *KCNQ3* is expressed in both axons and somatodendritic areas and to describe the potential differential roles.

*The changes are implemented (lines 78-84: “Together with *Kcnq2*, *Kcnq3* subunits are targeted to axon initial segments. Here *Kcnq3*-containing channels regulate the functional availability of Na^+ channels and affect spontaneous spiking, action potential amplitude and propagation^{18, 22, 23}. *Kcnq*/M-currents in somata and dendrites contribute to medium after-hyperpolarization, reduce excitability, prolong interspike intervals (ISIs) and regulate synaptic integration and subthreshold resonance²³⁻²⁷.”*

Page 3; dampening inhibition; this is unclear and should be expanded if there is space

The description has been extended by including “shunting inhibitory postsynaptic currents” (lines 85).

Line 122; without increased average firing rates of the mutant; They seem to be presenting numbers for the significant effect on bursts, but they should also present the numbers for the lack of significant effect on average firing rates; and be clear about which data address each point.

Now we report average rate of bursts, average firing rate (considering all spikes) as well as the rate of single spikes in extended new datasets in lines 132-139.

Line 132-133; This is a very confusing sentence that tries to merge two results about number of spikes and intraburst frequency. They should split this into a 2-3 sentences describing the results separately.

The sentence has been shortened: "In contrast to controls, in the mutant, longer bursts had higher frequency of spikes (i.e. shorter ISIs) than shorter bursts (Fig. 1f,h,i)." (lines 171-175)

Page 4 and Figure 1b,c legend (n=12 cells, n=11 cells). They do not seem to provide any statistical test for this difference in number and frequency of action potentials in slice preparations. They need to provide the statistical results for 1b and for 1c in the figure legend and/or the text.

Apologies for not giving a statistical evaluation. This is now detailed in the legend of the current Figures 1 d,e (lines 149-152).

Figure 1g; They show statistical significance within Kcnq3^{-/-} but they should present the comparison of the Kcnq3^{+/+} and ^{-/-} (black versus red). In general, they need to present their statistical results more clearly in all of their figure legends or text, as some statistical comparisons shown in the figures do not seem to be addressed in the text or figure legends.

We thank the reviewer for commenting on this unclear presentation (the comparison of groups was shown in the text (lines 128-129 in the previous version). We now show this comparison in the Figure 1i and provide further details in the Figure Legend and in the Supplementary Information (line 163, Supplement, lines 130-132).

Figure 2 legend; impaired readout (grey bar) of inputs from MS cholinergic neurons. The term does not seem at all appropriate for describing the modulatory activity of cholinergic neurons. Cholinergic modulation is instead probably modulating the input-output dynamics of hippocampal circuits. "Readout" should be changed to "modulatory effects" or something like that.

We agree that "readout" was not well chosen here and have changed it as follows: "impaired modulation (grey bar) of hippocampal (Hip) pyramidal cells (white triangle) via MS cholinergic neurons (white circle) in Kcnq3^{-/-} mice." Lines 191-192.

Line 176 - Page 6; 48 pairs; Why are there so few Kcnq3^{+/+} pairs?

We now performed additional recordings and analyzed previously recorded data from control mice with a genetic background identical to that of mutants. We obtained altogether 625 pairs of simultaneously recorded single units in Kcnq3^{+/+}. From this set

154 pairs (25% vs. 61% in *Kcnq3*^{+/+} vs. *Kcnq3*^{-/-}, $p < 0.0001$, χ^2 - test, lines 197-199) fired enough bursts during theta oscillations to compute cross-correlations. Similar to the initial findings in a smaller dataset, bursts in the extended control dataset were coordinated in the theta band (Figure 2 bc). Since the number of cell pairs still moderately differs between the groups, we tested whether the lack of theta-coordination in the mutant was due to a larger sample. This was not the case: the low coordination of bursts in the mutant reported in the Figure 2 was highly representative of the theta coordination in 100 randomly subsampled to $n=154$ subsets of *Kcnq3*^{-/-} cell pairs (Supplementary Statistics, lines 142-146).

Line 178; What is “Gaussian surprise”?

By surprise (please see Legendy and Salzman, J Neurophysiol., 1985, Gourevitch and Eggermont, J. Neurosci. Meth, 2007) we refer to a measure (computed here as SD above mean) of how unlikely it is that an event is a chance occurrence. The use of this infrequently applied term (aside from studies of spike trains, as $-\log P$ of a Poisson distribution) was indeed confusing. The probability of a spectral peak in a random distribution obtained by reshuffling is represented now as Gaussian percentile (Suppl. statistical information, Figure 2 bc). This permutation test is now included in main text (line 201) and explained in the Supplemental Statistics, lines 137-144).

Line 181 - Figure 2e; How can the *Kcnq3*^{-/-} be entrained to theta but not show theta rhythmic cross-correlations? This is confusing?

We thank the reviewer for pointing this out. The intact theta entrainment (Figure 2e) was observed only during immobility, whereas cross-correlations were computed for all theta rhythmic epochs, running as well as immobility. Computing cross-correlations separately for the two behaviors would require, in particular for alert immobility which is less frequently observed than running, substantially higher number of recorded cells to obtain sound number of pairs. We clarify in the legend to Figure 2 that the data used for cross-correlations were recorded mostly during running (when the entrainment is altered, Figure 2d, line 197).

Line 194;consistently high or more variable levels of acetylcholine; This section is confusing and should be split into more sentences that describe their point in more detail. In particular, they need to make clear that previous work showed that type 1 theta does not depend upon Ach and that type 2 does depend upon Ach and cite Kramis and Vanderwolf.

We have extended the description of the two types of theta, providing, as suggested by the reviewer, references to the initial study of the two rhythms and briefly highlighting their mechanisms and functional roles (lines 218-226): “The two types of theta oscillations were initially characterized based on the changes of the mesoscopic rhythm upon systemic application of muscarinic antagonists. These leave the power of local field potential (LFP) theta type 1 largely unaffected (atropine-resistant theta) while abolishing the type 2 (atropine-sensitive) rhythm⁴⁷. Atropine sensitivity of type 2 theta is owed to its reliance on the cholinergic excitation of PV-cells in the MS^{48, 49} which are crucial for theta rhythm. In line with cholinergic modulation of encoding new information³⁹, hippocampal acetylcholine levels during behaviors associated with type 1 theta are high⁵⁰. Behaviors connected to type 2 theta

are associated with more variable levels of cholinergic stimulation^{50, 51} which, similar to actions of acetylcholine in the cortex, may play a role in changes of information processing during increased attention³⁹.

Line 200 - Figure 2d; mutant was not modulated; This is confusing as the figure in 2d shows clear rhythmicity of the bursting relative to theta at double the frequency. They need to mention this and provide statistical tests showing lack of modulation to support this statement.

As suggested by the reviewer, we have now performed further statistical analysis comparing proportions and preferred phases of significantly modulated cells. The theta - modulated population in the mutant had a smaller size than in the control ($p < 0.0008$, χ^2 - test). These modulated cells were locked to broadly distributed phases resulting in a lack of the overall modulation at theta frequencies ($p = 0.14$, Rayleigh test). The distribution of the preferred theta phases in the mutant was indeed significantly bimodal ($p=0.032$, Silverman's bootstrap test), supporting the observed population rhythmicity at double the theta frequency. These results are now presented in the Suppl. Figure 3b and described in lines 232-242.

Line 203 Figure 2f; What is being shown in Figure 2f? Is it the difference between +/+ and -/-? If so, this needs to be much clearer as the figure legend makes it sound as if Figure 2f is only showing results from Kcnq3^{-/-} and does not mention that it is a difference between -/- and +/+.

Figures 2f does show the difference between Kcnq3^{+/+} and Kcnq3^{-/-}, sorry for the confusion. We have rephrased the legend to make this unambiguously clear (lines 209-211).

Line 224; display smaller place fields; Again there needs to be some presentation of statistical results to support this statement. The lack of statistical tests is surprising throughout the manuscript.

We thank the reviewer for this comment and agree that the graphical presentation of this key result was not sufficient. Statistical comparisons of place field sizes are now provided in lines 274-277.

Line 254; n=11 cells; why are there so few Kcnq3^{+/+} cells?

We have performed additional recordings in arena and circular track and now report results for more balanced datasets (arena/track: n = 76/78, Kcnq3^{+/+}, n = 95/94 cells, Kcnq3^{-/-}).

Line 257; inset curves; Are the insets on the same scale? What is the scale? Why is there a large increase at later times in the red plot for -/- This is surprising.

We apologize for the confusing presentation. We now provide a common scaling and numbers for the y-axis in insets which serve to easily appreciate the qualitative similarity of firing patterns. The plots show that in the mutant, the probability of single spikes, rather than being higher at longer latencies, is actually lower at shorter latencies, probably due to more vigorous bursting and hence more prominent

suppression of single spikes. The time course of activity changes is very similar between the genotypes as is shown now in properly formatted insets in Figure 3cd.

Throughout the figures they use the abbreviation (au); without definition. Presumably au means arbitrary units, but they should define this in the legends.

We have now defined a.u. in the legends of Figures 2 and 4.

Line 263 “which were no more phase-locked” do they mean “no longer phase locked”

Yes, corrected accordingly.

Line 278; citation 49; should this also include citation 46?

This reference was indeed confusing. The citation of Bender et al., 2015 (a paper from our own (AP) lab) refers not only to the statement about the experimental approach, but also to a small part of the dataset acquired for that paper and now analyzed in the present study (Figure 4c). To avoid any confusion, we also provide now the reference to Bender et al next to the description of Animals in Methods stating that “A part of the optogenetic stimulation dataset (from present PV-Cre mice) and of spontaneous neuronal activity recordings (in a wild-type mouse) have been used in previous reports^{28, 48}.” (line 600-602). The study of Dannenberg et al. is also cited in the context of similarity of the effects of axonal and somatic stimulation of MS PV-cells at theta frequencies (line 343-345).

Line 293 “the former synapse”. This is confusing as they are citing the paper by Robinson on glutamatergic inputs. Perhaps that paper did present results about the cholinergic input but maybe the phrasing of the sentence would be clearer without using the word.

Citations are now optimized as follows:

Line 409 “MS includes two interconnected GABA⁻ neuronal populations projecting to the hippocampus: cholinergic (ChAT⁺) and glutamatergic cells (Robinson et al). Cholinergic cells prominently synapse on CA1 area pyramidal cells (Frotscher and Leranth, 1985) and may therefore directly influence burst discharge.

Line 308; figure 4a “blue contours” where are the blue contours in Figure 4a? Do they mean in Figure 4b?

Apologies for the confusion (we did refer to Fig 4a): in Fig 4a changed to “blue cell/projections” (lines 366-367).

Line 313; red stripe; what are the blue bands in Figure 4b?

The blue bands depict light pulses, now clarified: “(light pulses - blue bars)” (line 370).

Lines 338-339 and discussion; I believe that tests of entrainment of theta by cholinergic versus GABAergic modulation of hippocampus were done in the Dannenberg paper 46. Probably what is different here is the focus on entrainment of bursts, but they need to be much clearer about this.

The focus of our study is indeed different from Dannenberg et al., 2015. We did not aim at comparing entrainment of various aspects of theta by cholinergic versus GABAergic inputs, but rather focused on the entrainment of bursts. The work of Dannenberg et al is discussed in lines 506-510 where we also clarify differences in objectives.

Line 373 “as well as the mediated via”. This is an awkward sentence structure and needs to be rewritten.

The sentence has been reformulated (lines 456-463): “Actions of acetylcholine on pyramidal cells include the inhibition of Kcnq3-containing M-channels, of slow AHP and of ,leak’ potassium currents^{20, 39, 57}. Muscarinic antagonists abolish burst firing in hippocampal pyramidal cells ex vivo and in vivo^{21, 58}. Conversely, reduction of M-currents in mice expressing a dominant negative mutant of Kcnq2²³ results in a higher number of spikes during bursts in neonatal mice⁵⁹. Postsynaptic M1 receptors also enhance Ih and Ca2+-dependent cation currents while presynaptic M4 receptors modulate Kir3 K+-channels and voltage-dependent Ca2+-channels³⁹. The latter effects of acetylcholine limit the spread of excitation in the CA3 region⁶⁰.”

Line 379 The link and citation regarding cholinergic modulation is appropriate but could be made clearer with more detail and more citations. (Hasselmo, 2006 provides a concise overview).

We thank the reviewer for pointing to this article. The discussion of cholinergic modulation of hippocampal information processing has been extended in lines 465-470.

Line 422; “unchanged LFP theta oscillations after ablation”; This did not seem to be shown explicitly in the figures or with statistical tests anywhere in the figures, the text or the supplemental materials. This comparison needs to be provided as both a figure and text.

Apologies for not presenting this result in sufficient detail. The earlier presentation included only an indication of a non-significant difference in the theta power in the legend of the Suppl Figure 2. We now provide details of the statistical comparison of the power of LFP theta oscillations in the legend of the Suppl. Figure 3c (Supplement, lines 48-49).

Line 512;a circular track; what is the symbol? Is that meant to be diameter? Should just use the word diameter

Changed to diameter.

Reviewer #3 (Remarks to the Author):

GENERAL COMMENT

The authors study how KCNQ3 channels affect spike bursts, single spikes, theta entrainment, and place fields in CA1 hippocampus. They employ an impressive variety of techniques: constitutive and conditional knockout mice, in vitro and in vivo electrophysiology, optogenetic stimulation and identification, and behavior, and the result is a set of interesting and potentially important observations. What is lacking, though, is a consistent through-line: a connecting theme to link the different parts of this paper together.

We thank the reviewer for these positive comments that appreciate the variety of techniques used and the importance of our results.

As it stands, the paper has two major parts: (1) a study of KCNQ3 knockout physiology (Figures 1-3) and (2) a study of how the different types of inputs from medial septum (MS) affect theta entrainment in CA1 (Figure 4). These two parts are not unrelated; M-type channels are blocked by acetylcholine and MS provides acetylcholine to CA1; but the connection seems weak. It is not clear how the results of part 1 should affect the reader's understanding of part 2 (and vice versa).

Apologies for not having been clear enough in explaining the logic of the study! We made numerous efforts to explain it better throughout the manuscript by

- introducing in the Abstract the optogenetic experiments as a tool to investigate upstream signals influencing Kcnq3-dependent burst firing (lines 46-48);*
- stating already in the Introduction that changes of bursts timing during theta oscillations in Kcnq3^{-/-} could be reproduced by optogenetic manipulations of MS afferents (lines 102-103);*
- introducing optogenetic studies of MS pathways potentially modulating burst firing and M-current (lines 332-337) and summarizing the optogenetic results accordingly (lines 435-438);*
- jointly discussing and connecting the theta entrainment of bursts in the mutant and in optogenetic experiments (lines 510-526).*

A related opinion: it is also not clear (to this reader, at least), how the major observation of Figure 2 (that theta coordination of spike bursts is disrupted after KCNQ3 knockout in moving mice) explains the major observation of Figure 3 (that single-spike place field sizes are smaller in the knockouts). Maybe the data of Figure 3c,d explain this, but if so, more explication would be appreciated.

This crucial point has now been further clarified in lines 294-309 and 327-329: "To investigate how interactions between bursts and single spikes at time scales of theta oscillations can influence spatial representations by the discharge rate we examined the influence of a burst on the probability of an ensuing single spike. In line with earlier reports^{1, 2}, firing of a single spike increased, and later decreased again, the probability of a subsequent burst (Suppl. Fig. 5c). Furthermore, in both genotypes, the probability of single spike firing following a burst was reduced during a period of up to 50-60 ms (Fig. 3c). These time windows of lower intrinsic excitability match the ascending part of the theta cycle when firing probabilities of pyramidal cells are overall

low due to the theta-rhythmic increase of inhibition. Accordingly, the recovery of intrinsic excitability after a burst coincides with times of increasing excitability during the theta cycle. The temporal match of an increased intrinsic and theta-oscillation related excitability can therefore facilitate the firing of single spikes during spatial navigation (Fig. 3c). Conversely, the timing of bursts which were no longer phase-locked during spatial navigation due to Kcnq3 ablation (Fig. 2) disrupted the temporal match of intrinsic and theta rhythm-driven excitability. Specifically, when bursts fired more often during descending phases in Kcnq3^{-/-} mice (Fig 2d, Suppl. Fig. 3b), the firing of single spikes fell on ascending theta phases associated with high inhibition. Hence, the firing of single spikes was reduced in Kcnq3^{-/-} mice (Fig. 3d) and this resulted in impaired spatial representations.”

SPECIFIC COMMENTS

(A) In vitro electrophysiology. The in vivo data of Figures 1-3 depend entirely on the full KCNQ3 knockout (the pyramidal cell-specific version is used only to make a small point). Given its importance, the authors should characterize CA1 pyramidal neuron intrinsic properties in this knockout mouse more thoroughly than they do. At present, the characterization is limited to injecting a family of current steps and measuring f-I curves. That is a good start; and the fact that they found a difference between control and knockout in this way is remarkable since an earlier study (Ref. 27) did not; but it is only a start.

To better characterize the cellular phenotype of the full Kcnq3 knockout we have characterized resting membrane potential and its changes during spikes, estimated mAHP and measured M-current in voltage-clamp recordings, as detailed further below. We agree that these are important parameters that were also requested by reviewer 1.

Some suggestions:

A1. Resting potential, resting input resistance, impedance. The authors suggest in the Discussion (lines 387-390) that the principal effect of eliminating KCNQ3 on intrinsic excitability is mediated by resting potential rather than medium afterhyperpolarization (mAHP). This might be true, but resting potential is not reported in this manuscript, nor is resting input resistance, and mAHP is not measured. While theta is not generated locally in CA1, entrainment to theta might be affected by the intrinsic resonance properties of CA1 neurons (see, e.g., Hu, Vervaeke, and Storm 2002), which might in turn depend on the M current. This issue might be probed, as Hu and colleagues do, by using a ZAP current (sinusoid of increasing frequency) to measure impedance.

Thank you for this comment, we have changed the manuscript accordingly. The measurements of the resting membrane potential and resting input resistance in pyramidal cells revealed no differences between Kcnq3^{-/-} or Kcnq3^{+/+} mice. We also measured the mAHP amplitude, including apamin-insensitive mAHP, and found no changes in Kcnq3^{-/-}. These results agree with the previous reports (Ref. 25,27), are mentioned in lines 122-124 and are shown in the Suppl. Figure 2. Together these results support neither an earlier proposed role of the resting potential, nor of mAHPs in the facilitated burst firing in Kcnq3^{-/-}. We discuss other possible mechanisms in lines

482-493 and point to the need of further investigations which go beyond the scope of our study.

We now consider in more detail a possible role of a subthreshold theta resonance as a mechanism contributing to the theta entrainment of bursts (lines 512 – 520):

*“The observed impaired rhythmic responses of pyramidal cells might be caused, for instance, by a failure to effectively follow theta rhythmic GABAergic inputs from MS. This scenario would be consistent with the contribution of M-currents to the subthreshold theta-band resonance in pyramidal cells^{23,75}. Yet we found that intact resonant properties of pyramidal cells, in wild type mice, may not be sufficient for the entrainment of bursts by GABAergic inputs alone. A more consistent explanation for the imprecise burst discharge of *Kcnq3*^{-/-} cells is their reduced capability to receive relevant timing signals in the absence of *Kcnq3*-containing M-channels.”*

A2. M current. The *KCNQ3*^{-/-} mouse does not lack M current in CA1 neurons because it continues to express *KCNQ2* (Supplementary Figure 1). As the authors note (lines 112-116) this; means that M currents are reduced in the knockout. It would be much more useful if they could show what the differences are, by directly measuring whole cell M currents in voltage clamp.

*Following the reviewer’s suggestion, we have measured M-currents in constitutive *Kcnq3* knockout and in controls. The mutant displayed a marked reduction of the M-current amplitude by approximately 50%. This important result is now reported in Fig 1 a,b.*

A3. If the authors do more in vitro experiments, they really should do so in older animals. The animals used here were so young (P15-20) that, not only is there a mismatch with the animals used in the in vivo experiments (12 weeks), but the other ionic mechanisms; that the authors postulate (lines 390-393) affect spike discharge are still developing.

To make voltage-clamp measurements more comparable with current-clamp data we have obtained earlier, we now performed new experiments and measured M-currents at ~P19, an age when neuronal excitability is already close to that of older mice.

(B) Lines 120-123. The authors note that average firing rates in vivo are not increased in the knockout. In fact, they are significantly decreased. Why is this point passed over without comment?

We thank the reviewer for making this point and now extended this result with an analysis of rates of single spikes and of bursts. Whereas burst rates are unchanged, the rate of single spikes is reduced, resulting in a reduction of overall firing rates. These mutually consistent effects are presented in lines 132-139.

(C) Movement vs immobility. The authors demonstrate in Figure 2d,e that theta coordination of spike bursts is abnormal in the knockout case when the animals are moving but not when they are immobile. This is a very interesting result, but I question this line: “In contrast, the population probability of burst firing in the mutant was not modulated by movement-related theta oscillations (Fig. 2d).” (Lines 199-201). To my eye, Figure 2d shows a frequency of twice theta: one peak on the decaying phase and

a second peak on the rising phase. These were population data (113 cells combined into a mean and SEM). It would be useful for the reader to know more about what the population distribution was. From Figure 2 alone, one could imagine a scenario where there were two ensembles of bursting cells that were theta-locked but asynchronous with each other. This scenario is unlikely, but I use it simply to argue that a fuller accounting of the population is in order.

The statement about population firing probability changes during movement – related theta was indeed confusing, sorry. We now modified the wording to stress the lack of the theta-frequency modulation rather than of any modulation:

*“Unlike the situation in WT mice, the population probability of burst firing in *Kcnq3*^{-/-} mice was not entrained by movement-related theta oscillations (Fig. 2d). (lines 230-232).*

We provide now additional measures of the entrainment – (1) fewer cells significantly entrained and (2) a uniform distribution of the preferred theta phases of the entrained cells - showing the lack of the burst discharge modulation at theta frequencies in the mutant (lines 232 – 237, Suppl. Figure 3b). Furthermore, the scenario outlined by the reviewer was correct – the distribution of preferred phases of individual cells in the mutants is indeed bimodal during spatial navigation (lines 238-239, Suppl. Figure 3b) with a fraction of cells preferentially firing on the decaying phase of the cycle. In contrast, during the immobility, pyramidal cells were locked close to theta troughs (lines 239-242).

(D) The spike burst abnormality of Figure 2 is not carried over into Figure 3, where only single spikes show a place field abnormality. As I noted above, I do not understand the relationship between the Figure 2 and Figure 3 results; or, more generally, what the authors mean when they write, as in the Abstract, about a balanced contribution of these firing modes in place fields. What does balance mean exactly?

We thank the reviewer for drawing attention to this point. We now provide a more detailed account of spatial representations, using a substantially extended dataset of place cell recordings acquired during the revision (mostly for the control group) (lines 278-284). While sizes of bursts place fields in the mutant were only slightly increased in the arena (Figure 3b), we found a marked increase of peak and average rates in place fields formed by spikes fired during bursts in the mutant (Table 2) which was observed in both types of enclosure. Further changes (in sparsity and spatial information) were found for spatial firing of spikes during bursts on the track (Table 2).

(E) The demonstration that, in mice, both GABAergic and cholinergic inputs from MS to CA1 are required for spike burst entrainment (Figure 4) is good. What would be better is if the authors could explain or even just speculate (in Discussion) why this is. Also, as noted above, it would be exceptionally good; for the sake of a through-line, if they could say quite what the results of Figure 4 have to do with the M current, which after all is the main subject of the paper.

We appreciate this suggestion of the reviewer. Due to its voltage dependence, M-current is probably most active at depolarized theta phases (likely in anti-phase with Ih-current), during which bursts are mostly emitted. The cholinergic inhibition of the M-current can therefore be essential for the peak excitability and bursting of pyramidal

cells at these phases. Simultaneously, the depolarization of pyramidal cells is also mediated by a perisomatic disinhibition, occurring when GABA-ergic MS inputs inhibit basket interneurons. These temporal interactions of cholinergic and GABA-ergic inputs may underlie their joint involvement in the entrainment of spike bursts. We provide these consideration in lines 527-539.

REVIEWERS' COMMENTS

Reviewer #1 (Remarks to the Author):

I appreciate authors' efforts to consider suggested corrections due to which the paper substantially improved. I accept authors' responses to my comments and happy it is proceeded for publication. The only thing to mention is that in Table 1 Excitability of pyramidal cells, the units of Depolarization rate during action potential and Repolarization rate during action potential are both strangely in mV/ms², whereas must be in mV/ms (or V/s), please correct. Also, please change the commas for full stop as decimal separator in the numerical values in the Table 1. It was a pleasure reviewing the paper.

Reviewer #2 (Remarks to the Author):

Overall, I am satisfied with the response of the authors to the reviewer comments. The revised article is much improved in many ways including enhanced clarity and inclusion of larger datasets. This article presents important results addressing the potential role of KCNQ3 subunits in the theta entrainment of burst firing and place field firing properties that give insights into its functional and clinically important role of this channel subunit.

I only have a few minor additional comments. The response to these comments could be evaluated by the editor.

Specific comments:

Line 43-44 – "...offset network oscillatory and intrinsic excitability." – The last part of this sentence in the abstract makes no sense to me even after reading the revised paper. This should perhaps be split into two sentences that make the meaning clearer.

Line 71 and before. "The translation of these input signals into location-specific hippocampal output involves regulation of excitability..." - This and the preceding two sentences are confusing because they imply that subcortical inputs are the main influence on place cells, instead of the important role of cortical inputs from areas such as entorhinal cortex. They should modify "these input signals" to make clear what they are referring to, and also clarify the source of the regulation of excitability. I assume they mean to say: "The translation of input signals from cortical structures into location-specific hippocampal output involves subcortical regulation of excitability..."

Line 132 – "number of spikes, in Kcnq3-/-" – the comma is unnecessary

Line 141 and 166 "accommodation...by approximately" – the text before and after Figure 1 does not seem to match up correctly.

Line 475 – "involved in retrieval67" – This added section is excellent and does not need revision, but the previous review was referring to the following article: Hasselmo (2006) Current Opinion in Neurobiology, rather Hasselmo and Giocomo, 2006.

Page 324 – "sinus wave" – should be "sine wave"

Figure 4 – Somewhere they should state clearly that Figure 4 does not present any results from Kcnq3-/- mice.

Signed by Michael Hasselmo

Reviewer #3 (Remarks to the Author):

The authors have done a good job of responding to the earlier comments. As a whole, the paper contains new and valuable data on how KCNQ3 channels affect bursts, spikes, theta entrainment, and place fields in CA1 hippocampus, and some interesting (and reasonable) interpretations. The

manuscript is well written and well organized.
No further comments.

Responses to referees' comments

Reviewer #1

I appreciate authors' efforts to consider suggested corrections due to which the paper substantially improved. I accept authors' responses to my comments and happy it is proceeded for publication. The only thing to mention is that in Table 1 Excitability of pyramidal cells, the units of Depolarization rate during action potential and Repolarization rate during action potential are both strangely in mV/ms², whereas must be in mV/ms (or V/s), please correct. Also, please change the commas for full stop as decimal separator in the numerical values in the Table 1. It was a pleasure reviewing the paper.

We appreciate reviewer's assessment of the revised manuscript. The typos have been corrected: the units in Table 1 have been modified to mV/ms and the commas replaced by full stops.

Reviewer #2

Overall, I am satisfied with the response of the authors to the reviewer comments. The revised article is much improved in many ways including enhanced clarity and inclusion of larger datasets. This article presents important results addressing the potential role of KCNQ3 subunits in the theta entrainment of burst firing and place field firing properties that give insights into its functional and clinically important role of this channel subunit. I only have a few minor additional comments. The response to these comments could be evaluated by the editor.

We thank the reviewer for a very positive feedback on the performed revisions and for the appreciation of the study relevance.

Specific comments:

Line 43-44 "...offset network oscillatory and intrinsic excitability" The last part of this sentence in the abstract makes no sense to me even after reading the revised paper. This should perhaps be split into two sentences that make the meaning clearer.

We have removed the ambiguous "offset" and reformulated the sentence as follows: "Less modulated bursts were followed by an intact post-burst pause of single spike firing, resulting in a temporal discoordination between network oscillatory and intrinsic excitability." (lines 43-45).

Line 71 and before. "The translation of these input signals into location-specific hippocampal output involves regulation of excitability" This and the preceding two sentences are confusing because they imply that subcortical inputs are the main influence on place cells, instead of the important role of cortical inputs from areas such as entorhinal cortex. They should modify "these input signals" to make clear what they are referring to, and also clarify the source of the regulation of excitability. I assume they mean to say: "The translation of input signals from cortical structures into location-specific hippocampal output involves subcortical regulation of excitability"

The sentence, presently in lines 72-75, has been modified as correctly proposed by the reviewer.

Line 132 "number of spikes, in Kcnq3-/-" the comma is unnecessary

Corrected.

Line 141 and 166 "accommodation" "by approximately" the text before and after Figure 1 does not seem to match up correctly.

The sentence interrupted by the Figure 1 "In control mice the initial three spikes of either short or long bursts were emitted in vivo with progressively increasing ISIs, resulting in frequency accommodation by approximately 20 Hz (Fig. 1f,h)." appears in the manuscript as intended.

Line 475 “involved in retrieval⁶⁷” This added section is excellent and does not need revision, but the previous review was referring to the following article: Hasselmo (2006) Current Opinion in Neurobiology, rather Hasselmo and Giocomo, 2006.

We have updated the reference 67 as suggested by the reviewer.

Page 324 “sinus wave” should be “sine wave”

Modified accordingly.

Figure 4 Somewhere they should state clearly that Figure 4 does not present any results from Kcnq3^{-/-} mice.

To clarify that optogenetic experiments were performed in mice with intact Kcnq3-containing channels we have modified the introduction to this part as follows: “To gain insights into the role of these hippocampal afferents in the theta entrainment of bursts, we recorded the discharge of putative CA1 pyramidal cells while optogenetically stimulating different inputs from MS in Kcnq3^{+/+} mice.” (line 272-274). The title of the Figure 4 has been also extended as requested by the reviewer: “Figure 4. Entrainment of pyramidal cell bursts in Kcnq3^{+/+} mice by collective rhythmicity of MS inputs.”

Reviewer #3

The authors have done a good job of responding to the earlier comments. As a whole, the paper contains new and valuable data on how KCNQ3 channels affect bursts, spikes, theta entrainment, and place fields in CA1 hippocampus, and some interesting (and reasonable) interpretations. The manuscript is well written and well organized. No further comments.

We are happy that the reviewer finds the data and their interpretations interesting and the revised manuscript well prepared.